# Quantifying gender biases towards politicians on Reddit

**Sara Marjanovic**[1], **Karolina Stańczak**[2]*, **Isabelle Augenstein**[2]

**1** Copenhagen Center for Social Science Data, University of Copenhagen, Copenhagen, Denmark,
**2** Department of Computer Science, University of Copenhagen, Copenhagen, Denmark

\* ks@di.ku.dk

**Data Availability Statement:** A minimal representation of the data-set (Reddit comment ID and linked Wikidata ID) is presented on GitHub due to file size constraints of Github, along with the code used (GitHub - spaidataiga/ RedditPoliticalBias: The data and code used in the

## Abstract

Despite attempts to increase gender parity in politics, global efforts have struggled to ensure equal female representation. This is likely tied to implicit gender biases against women in authority. In this work, we present a comprehensive study of gender biases that appear in online political discussion. To this end, we collect 10 million comments on Reddit in conversations *about* male and female politicians, which enables an exhaustive study of automatic gender bias detection. We address not only misogynistic language, but also other manifestations of bias, like benevolent sexism in the form of seemingly positive sentiment and dominance attributed to female politicians, or differences in descriptor attribution. Finally, we conduct a multi-faceted study of gender bias towards politicians investigating both linguistic and extra-linguistic cues. We assess 5 different types of gender bias, evaluating coverage, combinatorial, nominal, sentimental and lexical biases extant in social media language and discourse. Overall, we find that, contrary to previous research, coverage and sentiment biases suggest equal public interest in female politicians. Rather than overt hostile or benevolent sexism, the results of the nominal and lexical analyses suggest this interest is not as professional or respectful as that expressed about male politicians. Female politicians are often named by their first names and are described in relation to their body, clothing, or family; this is a treatment that is not similarly extended to men. On the now banned far-right subreddits, this disparity is greatest, though differences in gender biases still appear in the right and left-leaning subreddits. We release the curated dataset to the public for future studies.

## 1 Introduction

Recent years have induced a wave of female politicians entering office in the United States and Europe, including the first female US Vice-President (after 6 failed female presidential bids that year) [1, 2]. During the coronavirus pandemic, woman-led countries had significantly better outcomes (in terms of number of deaths) than comparable male-led countries [3].

However, women continue to be severely underrepresented in positions of power internationally, in what is called the "political gender gap" [4], due to the additional biases women encounter in politics. Both men and women prefer male leaders to female leaders, despite expressing egalitarian views [5, 6], and there is under-representation of women in all positions

production of my Master's thesis). However, the full data-set is available on the Harvard Dataverse (Replication Data for: Quantifying gender biases towards politicians on Reddit). These datasets are a subset of the already publicly available Pushshift dataset created using the official Reddit API with some additional Wikidata-sourced information regarding the mentioned politicians.

**Funding:** (IA) This work is partly funded by Independent Research Fund Denmark under grant agreement number 9130-00092B.

**Competing interests:** The authors have declared that no competing interests exist.

of authority [7, 8]. Given peoples' reported and implicitly measured aversion to female leaders [5, 6] and the reported effect of gender stereotypes on politician eligibility [9, 10], there is reason to believe specific biases may exist for female politicians.

We can detect expressions of biases by looking into patterns in text using methods from Natural Language Processing (NLP), the computational analysis of language data. Many studies on automatically detecting gender biases using supervised learning have focused on creating trained classifiers to detect misogynist or otherwise toxic language online to mixed success [11, 12]. Farrell et al. [13] measured the levels of misogynistic activity across various communities on the social media, Reddit, to show an overall increase in misogyny from the year 2015, even in female-dominated communities. However, biases can also present themselves in subtler manners. Therefore, there has been recent effort in releasing datasets and classifiers to detect condescension [14], microaggressions [15], and power frames [16]. Gender biases are also not limited to explicitly misogynistic language but can also appear as "benevolent sexism", which includes seemingly positive stereotypes about women [17], or as recurrent disinformation narratives and rumours spread about female public figures [18]. Therefore, it is important to look at language used in its context to determine biases.

Unsupervised techniques have been particularly powerful to identify themes in biased language. These uncovered biases could be unknown to even the text's author. Some pattern recognition methods used in NLP to uncover biases have been shown to mirror human implicit biases [19]. Initial NLP explorations of gender bias in text relied on word frequencies of pre-compiled lexica. Using pre-categorized word lists, Wagner et al. [20] first demonstrated the presence of gender biases in Wikipedia biographies—words corresponding to gender, relationships and families were significantly more likely to be found in female Wiki pages. Similar Wikipedia-centered studies found a greater amount of content related to sex and marriage in female biographies [21]. These probabilistic approaches rely on a bag-of-words assumption that fails to capture sentence structure and dependencies. To counter this, Fast et al. [22] extracted pronoun-verb pairs to compare differences in the frequently linked adjective and verbs across gender in online fiction writing communities; regardless of author gender, female characters were more likely to be described with words with weak, submissive, or childish connotations. Similarly, Rudinger et al. [23] used co-occurence measures to showcase biases across gender, age and ethnicity via the top word co-occurrences in paired image captions. They found clear stereotypical patterns that cut across both age and ethnicity—women-related words were associated with emotional labour and appearance-related words as well as their male relations. Ultimately, across literature, news, and social media, there is a consistent pattern of women being described in terms of their appearance, emotionality, and relations to men. In contrast, men are described more in terms of their occupation and skill [24]. These are the subtle ways in which biases can manifest in daily language.

However, biases can also be found when examining simpler surface cues. We define these differences as 'extra-linguistic cues'. For example, online text about women is consistently shorter and less edited than corresponding texts about men [25, 26]. In addition, Wikipedia articles about women are more likely to link to men than in the opposite direction [20] and articles about male figures are more central [21]. In our study, we use a variety of methods to analyse both linguistic (e.g. in terms of language used) and extra-linguistic cues (e.g. figure centrality) presenting a comprehensive study of hostile and benevolent gender biases towards politicians in society.

Within social media, Field and Tsvetkov [27] find that comments addressed to female public figures could be identified by the prominence of appearance-related and sexual words, echoing the findings of general gender bias studies outlined above. Comments addressed to female politicians, however, are harder to identify; the terms most influential to their

identification are related to strength and domesticity. However, the model they trained on comments addressed to male and female politicians still achieved above-chance performance in identifying microaggressions without any overt indicators of gender. Despite similar tweets by male and female politicians, tweets addressed *towards* politicians differ greatest along the gender axis; with female politicians targeted with more personal than professional language [28]. However, all of these studies on political gender biases have relied on messages addressed *to* politicians. In contrast, in our study we look at conversations *about* male and female politicians.

We continue to explore these systematic differences in female portrayal by centering on discussion about male and female politicians on online fora, which provides a different presentation of biases in public interest and social expectation than previously examined media [29, 30]. While gender biases have been shown to differ across languages and cultures [20], we focus on gender biases in English given its predominance online. We expand the cultural relevance of this study outside of exclusively United States by explicitly taking comments from other English-speaking communities (such as Canadian, Australian and Indian-specific online communities).

In this work, we focus on three main **contributions**. First, we curate a dataset with a total of 10 million Reddit comments which enables a broad measure of gender bias on Reddit and on partisan-affiliated subreddits, which we make public for future investigations. Second, we do not merely analyse for hostile biases, but assess also more nuanced gender biases, i.e. benevolent sexism. Finally, we quantify several different types of gender biases extant in social media language and discourse.

We rely on both extra-linguistic and linguistic cues to identify biases. The extra-linguistic analyses look at differences in the amount of interest devoted to politicians (**coverage biases**) as well as how these politicians are related to one another in comments (**combinatorial biases**). We also look at linguistic biases; these include differences in how public figures are named (**nominal biases**), attributed sentiment (**sentimental biases**), and descriptors used (**lexical biases**). Through this examination, we also compare how these biases are presented across different splits of the dataset to show how biases can differ across political communities (left, right and alt-right). The investigations allow us to comprehensively measure the manifestations of biases in the dataset, forming a reflection of what biases are present in public opinion.

## 2 Data

We consider our curated dataset as one of the main contributions of this study. We make the comments publicly available for use in future studies (https://github.com/spaidataiga/RedditPoliticalBias).

Many related studies on gendered language have relied on large corpora [24, 31–33] or data collected from Twitter or Facebook [27, 34], but the structure of these media as massively open fora limit researchers' ability to compare language use across community and context. We rely on a different social media platform: Reddit. Reddit is divided into various sub-communities known as 'subreddits' (denoted by the prefix /r/). Subreddits reflect different areas of interest users can choose to engage in, which could be related to the community's location, hobbies, or overarching ideology. These subreddits are moderated by volunteer members of the community. The moderation within these ecosystems can be seen as standards that reflect acceptable conversation within each community, all of which contain their own norms [35]. This ecosystem has previously been used to study the effect of online rule enforcement [36] and its effect on hate speech [13, 37].

A two-year time period between the years 2018—2020 (exclusive) is selected for data collection. The two-year length is chosen to mitigate confounds due to seasonal and topical events. This specific time period is selected for two reasons: Due to the record-breaking number of women elected in the 2018 US Midterm Elections, many of whom are women of colour or other minority status, 2018 has colloquially been named "The Year of the Women." [1]. This allows for the perfect opportunity for an investigation of the language used in gendered political discussion, as it would be less skewed by specific prominent individuals. Data collection ends at the start of 2020 due to the change in Reddit's content policy in June 2020 [38] that led to the banning of several fringe subreddits, including /r/the_Donald, which is included in the data collection.

We draw our Reddit data from the Pushshift dataset, a 7-year-old publicly available Reddit archive queried using the official Reddit API and curated for use in scientific research [39].

To provide sufficient context for the NLP library tools, we restrict the two-year comment dataset to only comments with an entire conversational context. As posts are archived after 6 months, each comment can have a maximum 6-month long conversation history. Therefore, we only look at comments between the time period July 1 2018 00:00:00 GMT through to December 31 2019 23:59:59 GMT.

A 2016 survey of Reddit users finds the general user base is predominantly young, male and white. This skew is stronger in larger subreddits, such as /r/news [40]. However, different subreddits can be expected to have different distributions of user age, ethnicity, political orientation, and gender. By comparing the language used across subreddits, one can see the different standards for acceptable conversation across these communities. We selected to scrape from a collection of relatively popular, active subreddits that we predicted to have a more diverse audience and be more likely to contain political discussion.

The overwhelmingly popular /r/news and /r/politics are expected to generate high amounts of political discussion. However, other subreddits are selected to facilitate possible comparisons of interest and to diversify the dataset in terms of poster political alignment, country of origin, gender, and age.

Reddit is predominantly left-leaning, with less than 19% of overall users leaning right [40]. Since the larger subreddits, /r/news and /r/politics, therefore, can be expected to have discussion that reflects centre-left perspectives, we scrape from explicitly partisan subreddits to expand the versatility of our database. We separate the alt-right community in the subreddit /r/the_Donald from other right-leaning communities given its controversy on the platform [38] and within the US Republican community(See: https://rvat.org/).

We also expand the global representation of the database in the selection of subreddits as 50% of the general Reddit userbase comes from the United States. We locate subreddits specific to English-speaking countries for collection. Subreddits of other languages would be interesting, however, they are excluded given the relative lack of language processing resources for the countries in question (in particular, co-reference resolution and entity linking).

We also include data from subreddits that are expected to have gendered, though not necessarily political, discussion. /r/TwoXChromosomes and /r/feminisms are two female-centric subreddits that are expected to contain more female-positive perspectives than other more male-oriented to gender-neutral communities. Likewise, /r/MensRights, a male-oriented subreddit criticised for misogynist tendencies [13], is expected to use different language in gendered political discussion. Finally, to broaden the age range of the dataset, we also scrape from /r/teenagers, a subreddit geared towards youth, to include discussion generated from a presumably younger population than the aforementioned subreddits.

Wikidata is used to collect an exhaustive list of international male and female politicians. Though their collection of all elected political officials is not complete, Wikidata reports fairly

high (over 95%) gender coverage of politicians across most countries [41]. This query obtained data for 316,743 political entities (259,165 cis-male, 57,502 cis-female, and 76 entities outside of the cisgender binary). While it is relevant and interesting to explore how results differ across the gender spectrum, we restrict this investigation to politicians within the cis-female and cis-male binary, given the low prevalence of gender-diverse politicians in the dataset.

Firstly, coreferences within the comments contained within the dataset are resolved using the HuggingFace neural coreference resolution package (available on https://github.com/huggingface/neuralcoref). To identify the politician discussed in each post, a state-of-the-art lightweight entity linker (REL) [42] is used to mark each comment with the associated wiki-data ID. This was selected after a comparison of four different state-of-the-art entity linkers on a manually labelled dataset of 100 comments; however, it should be noted that the correct female entity is only caught in 50% of the labelled cases. Therefore, it is highly likely that many comments discussing female politicians are missed in this dataset. As REL maps to Wikipedia pages, these are then translated to Wikidata IDs using the python library wikimapper (https://pypi.org/project/wikimapper/).

Only comments directly discussing a known politician are kept in this dataset (either via use of a name or coreferent). The referent for each politician is replaced by the token [NAME] and the text is saved for analysis. For every entity mentioned in a comment, a data point is made. Therefore, some comments contribute to multiple data points. Extrapolating from the observed accuracy of the context coreference resolution and entity-linker along the pipeline, it can be estimated that approximately 31.0% of the political conversation about women in the selected subreddits is captured in this pipeline.

To mitigate any confounding effects of automatic posters, we remove all comments made by bots by searching for a bot-related disclaimer on each post relying on the subreddit moderation.

This leaves a final dataset of 13,795,685 data points (where each data point corresponds to one politician mentioned). Within this dataset, 8,170,625 comments mention one single entity (7,190,082 male; 980,543 female). The remainder of the data points correspond to 2,317,117 comments mentioning two or more political entities (4,815,262 male; 809,175 female). There-fore, this dataset stems from a total of 10,487,742 unique comments. The average comment is 51.28 ± 65.43 tokens long. 19,877 different political entities are mentioned in the dataset (16,135 male; 3,742 female). These politicians come from 312 different lands of origin (as determined from their WikiData properties), but the vast majority of comments (88.89%) refer to politicians born in the United States.

The final dataset includes comments from 24 subreddits. Most comments (70.63%) come from the subreddit /r/politics. 425,472 comments come from subreddits expected to be left-leaning. 420,895 comments come from right-leaning subreddits. and 1,664,335 comments come from the subreddit /r/the_Donald (which is from now on described as the "alt-right" group and is separated from the right-leaning subreddits given its already outlined controversy within the Republican community and Reddit). Therefore, 2,510,702 comments come from explicitly partisan communities. All selected subreddits are listed in S1 Table in §6 alongside the number of comments and their partisan-affiliation (if any).

## 3 Analyses

Gender biases can be assessed in a variety of different methods to reveal different types of bias. In this work, we analyse linguistic and extra-linguistic cues to broadly investigate gender bias towards politicians. To this end, we employ several different methods within extra-linguistic (§3.1) and linguistic analysis (§3.2) that we introduce below. With each investigated bias, the

hypothesis phenomenon is first defined, followed by the methodology used in its assessment. We also showcase the applicability of our dataset to inter-community comparisons by conducting the same comparisons on partisan subsets of the data (which include only explicitly left, right and alt-right leaning subreddits).

### 3.1 Extra-linguistic analysis

As gender biases can be measured without looking at the actual content of a document, we first explore "extra-linguistic" biases within comments, in terms of public interest in politicians and how politicians are discussed in the context of other politicians.

**3.1.1 Coverage biases.** *When taking into consideration the numbers of male and female politicians, do online posters display equal interest?*

We answer this question by comparing the relative coverage of male and female politicians. Coverage biases are a staple of many gender bias investigations [20, 26, 43] and can be assessed in a myriad of methods: the percentage of comments about men/women, the proportional number of politicians discussed, the amount of comment activity generated about each politician, and the amount of text in each data-point.

To assess the number of politicians discussed, it is vital to consider the disparity in number of male and female politicians. Women, internationally, are significantly less likely to hold positions of office [4]. Even were a 50% parity of male and female politicians to exist in the near future, the historical lack of female political figures ensures a significant disparity in the possible number of male and female politicians in popular discussion. Therefore, we look at the proportion of male and female political entities extracted from Wikidata present in the dataset. This carries the assumption that Wikidata can be used as a "gold standard" of politician coverage, as it could hold biases of its own and does not have complete coverage of all politicians.

In addition, we also look at the number of comments generated about each politician entity (the politician's "in-degree"). The distribution of in-degrees are then compared using the two-sample Kolmogorov–Smirnov test [44], a non-parametric test that assesses for a significant difference in two distributions. Given that some politicians obtain considerably more attention than others (e.g. Donald Trump is mentioned in 3,208,707 comments.), normal parametric statistical metrics would not be suitable for such a skewed distribution. We report the D-values and use a critical value of .01 to determine significance.

Finally, we look at the amount of text in each comment as a measure of the degree of activity, per unit of activity. We compare comment text length as determined by the number of tokens in each comment body. This investigation is isolated to only comments describing a single politician. We compare for significant differences in the distribution of comment lengths describing male and female politicians using student t-tests with a critical value of .01. We report Cohen's D as a measure of the effect size [45].

When looking across the political spectrum, we rely on two-way ANOVAs to assess for significant main effects in gender and partisan divide, as well as interactions between the two. ANOVA tests are conducted in R with posthoc Tukey-HSD tests to determine significant pairwise differences.

**3.1.2 Combinatorial biases.** *When female politicians are mentioned, are they mentioned in the context of other women? Or as a token women in a room of men?*

We assess for combinatorial biases that appear in the discussion of multiple political entities, following Wagner et al's work in uncovering structural bias in Wikipedia article linkage [20]. This is accomplished through the measure of gender assortativity (the tendency of an entity of one gender to be linked to another of the same gender). These acted as measures of

the "Smurfette principle", which posits that women are more often found in popular culture as peripheral figures in a network with a core comprised of men [46]. In this investigation, we look at comments that mention multiple politicians and compare the conditional probability $L(g_{additional}|\exists g_{given})$ that a comment will mention an entity of gender $g_{additional} \in \{female, male\}$, given at least one mention of $g_{given} \in \{female, male\}$.

Unlike Wikipedia pages and links, which can be approximated as a directed graph, comments describing one or more politicians can not be so easily approximated with pairwise relations. Proper modelling of these polyadic relations would require the computation of Higher-Order Networks [47]. However, even when homophilous preferences should be expected to be present in a dataset, it has been shown to be combinatorially impossible to express two simultaneous homophilous preferences with higher-order networks [48], as had been explored in similar studies of gender biases [20].

To approximate these measures, we calculate the conditional distribution $L(g_{additional}|\exists g_{given})$ using. Eq 1, which carries the caveat that, should $g_{additional} = g_{given}$, $\Sigma(g_{additional}|\exists g_{given})$ does not include the pre-existing mention of $g_{given}$ in each comment. Therefore, should a comment mention just one female politician, $\Sigma(female|\exists female) = 0$. For a comment mentioning two female politicians, $\Sigma(female|\exists female) = 1$.

$$L(g_{given}, g_{additional}) = \frac{\sum_{i=0}^{N}(g_{additional}|\exists g_{given})}{\sum_{i=0}^{N}(\exists g_{given})} \tag{1}$$

However, this approximation does not take into account the relative prominence of both genders in the dataset, where male politicians are significantly over-represented. While this is certainly an example of a bias in elected politicians as well as a bias in political discussion (e.g. coverage bias), it does necessarily reflect a combinatorial bias. Ignoring this issue would bias the conditional probability to over-estimate a comments likelihood to link to a male politician. In Wagner's study of article assortativity [20], the conditional probability was scaled by the marginal probability of the linked gender from the article's gender. However, given the undirected nature of these associations and the limitation of this dataset to two genders, adjusting for the prominence of the "additional" entity's gender makes evaluation of homophily impossible. The obtained values in this study can only be compared when $g_{additional}$ is shared, given that the values then share the same marginal probability.

Therefore, to account for the disparity in the representation of male and female political entities in the dataset, we create a null distribution, a powerful, yet simple simulation technique that allows significance testing of an observed value. We create $10^5$ null models on the data set and compute the resulting value from Eq 1 to create the null distribution. A critical value of.01 is used to determine significance.

## 3.2 Linguistic analysis

The remainder of the biases examined in this corpus investigates the actual language used in the political discussions. We investigate differences in how politicians are named, the feelings expressed about politicians, and the senses of the words used.

**3.2.1 Nominal biases.** *Do people give equal respect in the names they use to refer to male and female politicians?*

Scholars have noted differences how male and female professionals are addressed; Women are exceedingly named using familiar terms, thereby lowering their perceived credibility [49,

50]. We investigate this phenomenon by comparing the name used in reference to the political entity with the linked entity's recorded name data (as accessed from Wikidata).

If a first or last name is not provided for the entity in question, the names are approximated by splitting the politician's full name across spaces. In the extracted Wikidata dataset, first names are recorded for 82.9% of entities, and last names are recorded for 56.7%. A politician's first name is approximated to be the characters before the first space in their full name. The last name is approximated to be all characters following the final space in their first name. This approximation is not ideal as there are many cultural variations in first and last name presentation. Some cultures may have first or last names that are space-separated, and many Asian cultures flip the order of the given name and surname in the full name. However, given the dataset's bias towards Western politicians, we expect limited noise from this assumption. We then compare the usage of these names across politician gender.

Calculation of referential biases in this manner also ignores any politicians that are regularly referenced using a nickname (e.g. 'Bernie' for U.S. Senator Bernard Sanders, or 'AOC' for U.S. House Representative Alexandria Ocasio-Cortez). Given the low availability of this information on Wikidata and the lack of other resources, we do not compile a list of common nicknames for politicians, as it would require manual research into all politicians to create an exhaustive list of known nicknames. References outside of the list of expected names for the entity are saved as 'Other' and include nicknames as well as misspellings of the politician's name (e.g. 'Mette Fredereksin' for the Danish Prime Minister Mette Frederiksen).

Given that these are categorical variables, we rely on the chi-square test [51] to determine a significant difference in frequencies of name use across gender by comparing expected frequencies. The strength of these associations is given by Cramer's V [52]. For pairwise comparisons of interest, we calculate odds-ratios, a measure of association between two properties in a population. Odds-ratios are reported with a 95% confidence interval. A confidence interval exclusive of the value 1.0 suggests significance with a critical value of .05.

In the cross-partisan analysis, we rely on log-linear analyses to assess significant differences in category proportions to find the most parsimonious model that significantly fits the data (as assessed via a likelihood-ratio test). Depending on the complexity of the final model, it is then further analysed along two-way and one-way interactions using chi-square tests and Cramer's V, followed by odds-ratio comparisons.

**3.2.2 Sentimental biases.** *When people discuss male and female politicians, do they express equal sentiment and power levels in the words chosen?*

Previous gender bias studies have shown higher sentiment towards female subjects [22, 33, 53] (in what we have previously described as 'benevolent sexism'); however, there is evidence that this finding varies along the political spectrum [28]. In addition, studies on film scripts and fanfiction have shown lower power and dominance levels for female characters [22, 54]. We are interested in exploring these biases in the political sphere, given people's known predispositions against female authority [5, 6]. To accomplish this, we rely on the NRC VAD Lexicon [55], which contains over 20,000 common English words manually scored from 0 to 1 on their valence, arousal and dominance. For example, the word 'kill' is scored with a valence of .052, and a dominance level of .736. In contrast, the word 'loved' has a valence of 1.0 and a dominance score of .673. At its time of publication, it was by far the largest and most reliable affect lexicon [55], and it continues to be widely used in linguistic studies [33, 56, 57].

We rely on the valence and dominance ratings of this lexicon to calculate the sentiment and perceived power levels of comments describing singular politicians (to allow for easier sentiment attribution). For every comment, the body text is converted to lower-case, but not lemmatized given the lexicon's inclusion of different word morphologies. The valence and dominance score of each in-corpus word in the text is summated and averaged over the text's

number of in-corpus words to determine the comment's average valence and dominance. The score is averaged over in-corpus word count as other studies have found that only a portion of words in a text can be expected to be covered [33, 57]; this coverage is expected to be even smaller on social media text. The calculation of these averages is followed by student t-tests to detect statistical significance with a critical value of.01. In the case of cross-partisan analysis, two-way ANOVAs are performed, followed by post-hoc Tukey HSD tests [58], to find significant differences in means of valence and dominance scores for male and female politicians. Cohen's D is used as a measure of effect size.

To compound this test, we also measure the output of a state-of-the-art pretrained sentiment classifier on the comment text. We rely on a RoBERTa-based sentiment classifier that outputs a positive or negative sentiment label to a maximum 512 token input text [59]. We chose this model due to its high reliability across data-sets and tolerance for long input. The authors report its 93.2% average accuracy across 15 evaluation data-sets (each extracted from different text sources). We then assess for a significant difference in the categorical output of this model across genders using a chi-square test. Cramer's V is used to measure the strength of the association. Cross-partisan analyses use log-linear analyses to find the most parsimonious model. The final model is then analysed further using chi-square tests, Cramer's V, and odds-ratio comparisons, depending on the significance and strength of the associations.

**3.2.3 Lexical biases.** *When people discuss male and female politicians, how do the words they use to describe them differ?*

We show gender biases in word choice in political discussion using point-wise mutual information (PMI), a statistical association technique originally derived from information theory [60] which has been used to show gender biases in image captions, novels, and language models [23, 31, 61]. PMI is computationally inexpensive and transparent, since it allows for significance testing, intuitive comparisons along contexts, and confound control with small modifications [62, 63]. The isolated most 'gender-biased' words can then be analysed more deeply, either manually or with the use of pre-labeled lexica.

In this study, we investigate the co-occurrence of words with a politician's gender. We take descriptors linked to a political entity and calculate the probability of their co-occurrence to a gender across entity. These descriptors are obtained through a dependency parsing pipeline as nouns, adjectives, and adverbs parsed as children of the entity in question. The use of dependency-parsed descriptors also allow for the analysis of comments discussing more than one political entity. We then count the frequency of the lemmas of these descriptors across gender. Words with high PMI values for one gender are then suggested to have a high gender bias.

$$PMI(x, y) = ln\left(\frac{P(x, y)}{P(x)P(y)}\right) \tag{2}$$

One issue that arises from this method is that words that are particularly linked to one popular politician may then be confounded as linked to that politician's gender. For example, a prominent Somalian female politician may cause the word 'somalian' to be inappropriately linked to the gender 'female', though both men and women should be equally likely to be described as Somalian as it is not an innately gendered word. Therefore, we follow the lead of Damani by calculating PMI from a document count, not total word count. These additions allowed his PMI measure to give closer to human-labelled results on free association and semantic relatedness tasks in nearly all tested large datasets [62].

The original cPMId looks at significant word co-occurrences using document counts rather than word counts. Documents within a corpus are counted as containing a significant word co-occurrence if the co-occurrence surpasses a threshold of word co-occurrence within a

pre-determined word span parameter. The frequency of documents containing the co-occur-rence of items x and y are represented as $d(x, y)$ across the number of documents $D$. However, we are not looking at the co-occurrence of words within a word span, but in the co-occurrence of a word with a gender across a range of entities. Therefore, we view each political entity as a document and count a word's usage across different entities. Usage of this final Eq 3 is expected to minimize the appearance of obviously confounding descriptors in the top female and male word lists. For both cases, to limit PMI's potential to bias towards uncommon words, we only consider words that had a minimum count of 3 for both genders.

$$PMIe(word = w, gender = g) = ln\left(\frac{e(w,g)}{\frac{e(w)e(g)}{E}}\right) \tag{3}$$

Next, we analyse the top 100 gendered attributes for both male and female politicians. Within these words, we can assess for patterns in word types, vulgarity or even sentiment (thereby, further investigating the dataset for instances of hostile or beneveolent sexism). Simi-lar studies [31] have mapped pre-made word senses to the obtained words to visualize biases. Given the informal source of the dataset, we do not expect that many of the words extracted to be present in many existing resources.

We enlist the help of two volunteer annotators to label the obtained words using pre-defined labels. The volunteers enlisted are young (below 35), trained in the social sciences, and familiar with the platform Reddit. Volunteers are asked to mark each word with a hand-coded sentiment ranking (Negative, Neutral or Positive) and to label each word as belonging to one of the following 8 categories, compounded from findings in prior literature on the subject (our specific motivations for the inclusion of each sense are further outlined below):

- **Profession**: A term related to someone's profession or work activities (e.g. politician, speaker).

- **Belief**: A word relating to a politician's political ideals (e.g. republican, antifa)

- **Attribute**: A word related to a politician's supraphysical attributes (e.g. intelligent, rude)

- **Body**: A word related to their body (either a body part or general attractiveness/sexuality) (e.g. nose, beautiful)

- **Family**: A word related to a politician's family (e.g. mother) or relations with others (e.g. lover) or (in)capacity for that. (e.g. childless, pregnant)

- **Clothing**: A word relating to clothing/fashion/attire (e.g. fashionable, suit)

- **Label**: A general metaphor/term applied to someone (i.e. 'name calling') that may not fit neatly into one of the above categories (e.g. bitch, angel)

- **Other**: A word that doesn't fit into any of the above categories (e.g. phone, song)

We assess the traditional senses of Profession, Family and Appearance, given a wealth of extant NLP studies investigating gender biases that show greater job-relevant language attrib-uted to men [20, 24, 28], greater information about personal-life (i.e. family and relationships) and language about appearance in text about women [20, 23, 24, 27, 31, 33, 64, 65]. Popular articles comparing male and female politicians suggest similar issues [66], but add in addi-tional concerns: Female politicians are held to a higher set of standards than men. They must be likeable; unlike men, they cannot be shrill, outdated, or flawed [67–69]. In addition, their choices of outfit often are central in their media attention [70, 71]. Therefore, we also look

specifically at words describing a politicians clothing. We also separately mark politically-relevant ideals (Belief) and other metaphysical characteristics (Attribute). Finally, we include the final category "Label" to include sentimental differences in other terms or metaphors used to describe politicians.

For the general dataset, a chi-square test of independence with a critical value of .05 is performed to assess a significant relationship between politician gender and sense distribution. We choose a less-conservative critical value for this investigation given the smaller sample size relative to the other analyses. Cramer's V is calculated as a measure of effect size. Pairwise comparisons of interest are made using odds-ratio calculations.

For cross-partisan analyses, given the smaller datasets, only the top 50 gender-biased words for each gender and partisan-group are extracted for annotation. This is to conserve annotator time and to ensure that the list for the most "male"-skewed words are reliably male-skewed (we find fewer male-biased words, especially on the less-popular subreddits). However, given the large number of labels and fewer annotated samples, the total sample size does not meet the minimum requirements for log-linear analysis [72]. Therefore, these results are presented graphically and are only compared using odds-ratio comparisons.

# 4 Results

## 4.1 Coverage biases

*When taking into consideration the numbers of male and female politicians, do online posters display equal interest?*

We find equal coverage of male and female politicians across number of political entities mentioned, activity generated per politician, and length of text discussing each politician.

Though comments about female politicians make up only 16.09% of the data points, 6.23% of male political entities available from WikiData are mentioned in the dataset, and 6.51% of female WikiData political entities are mentioned.

We present the distribution of politician in-degree across gender as a complementary cumulative distribution function in Fig 1. The overlap of the two distributions suggests that male and female political entities do not differ in the activity generated per politician (despite fewer comments about female politicians). Though the tail of in-degree for male entities is longer, it likely corresponds to one or two notable male entities (i.e. former or current presidents). In addition, Kolmogorov-Smirnov tests do not find significant differences in the two distributions ($D = 0.017$, $p > .05$).

Student t-tests find a significant difference ($t(13795060) = 27.16$, $p-value < .0001$) between the lengths of comments discussing male ($40.19 \pm 37.55$ tokens) and female politicians ($34.23 \pm 34.09$ tokens). However, the effect size of this difference (Cohen's D: 0.16) is negligible.

These results suggest that male and female politicians receive equal portion of comments. This finding contradicts with previous research showing greater coverage of male politicians in media [43]. However, unlike this investigation, these studies analyse media attention rather than public interest.

**4.1.1 Cross-partisan comparison.** We find that female politicians receive smaller public interest in all partisan divides of the subreddits. However, left-leaning subreddits show the most equal coverage of male and female politicians. Alt-right discussions contain significantly more comments about male politicians.

When looking exclusively at the partisan subset of the data, we find a smaller portion of comments discussing female politicians in all partisan divides of the subreddits than in the general Reddit conversation. Despite this, left-leaning and alt-right subreddits show relatively

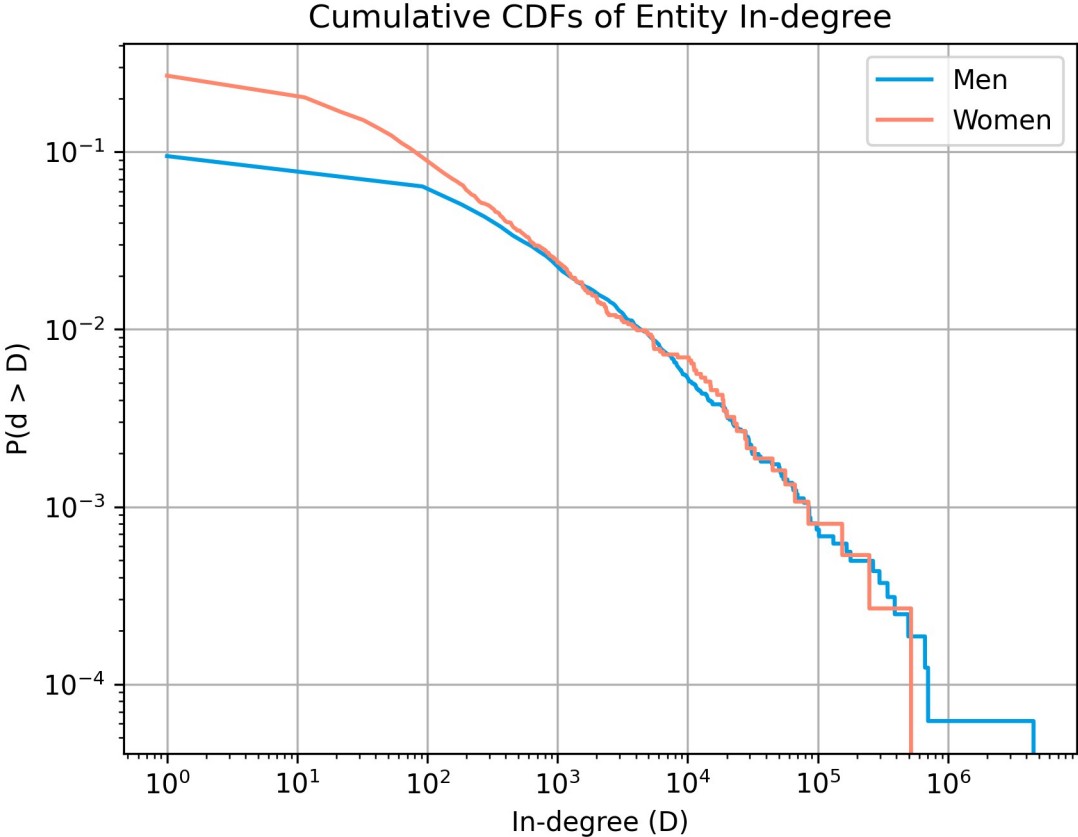

**Fig 1. The complementary cumulative distribution function of the in-degree distributions of male and female political entities.** Politician in-degree distributions.

more comments discussing female politicians (15.1% and 15.7% of comments, respective to each divide) than that is seen in the right-leaning subreddits (12.4% of comments).

In terms of the number of political entities mentioned, there are fewer entities mentioned in all divides than in the general dataset. However, the overall proportion of male and female politicians is relatively equal across divides. On the left-leaning subreddits, 1.56% of collected female politicians and 1.56% of collected male politicians are mentioned. On the right-leaning subreddits, 1.04% of collected female politicans and 1.09% of collected male politicians are mentioned. 1.85% of female politicians and 1.87% of male politicians seen on the Wikidata database are mentioned on the alt-right subreddit, /r/the_Donald.

Looking at the plot of the distribution of comments per political entity in Fig 2, there is a similar distribution across gender in both the left and right-leaning subreddits. However, there is a significant difference between the genders when looking at the alt-right subreddit, /r/the_Donald. More comments are consistently made about male politicians than female politicians, suggesting male politicians have greater centrality in alt-right political discussions. The results of Kolmogorov-Smirnov tests are reported in Table 1.

Finally, when looking at comment lengths, two-way ANOVAs show significant main effects of partisanship ($F(2, 1598999) = 11858.9; p < .0001$) and politician gender ($F(1, 1598999) = 6321.8; p < .0001$), as well as a significant interaction ($F(1, 1598999) = 172.2; p < .0001$). Post-hoc Tukey HSD tests show that comments about men ($\mu = 42.8 \pm 61.7$) are consistently longer than those about women ($\mu = 31.7 \pm 46.0$)($p < .0001; d = .19$). Comments on right-leaning

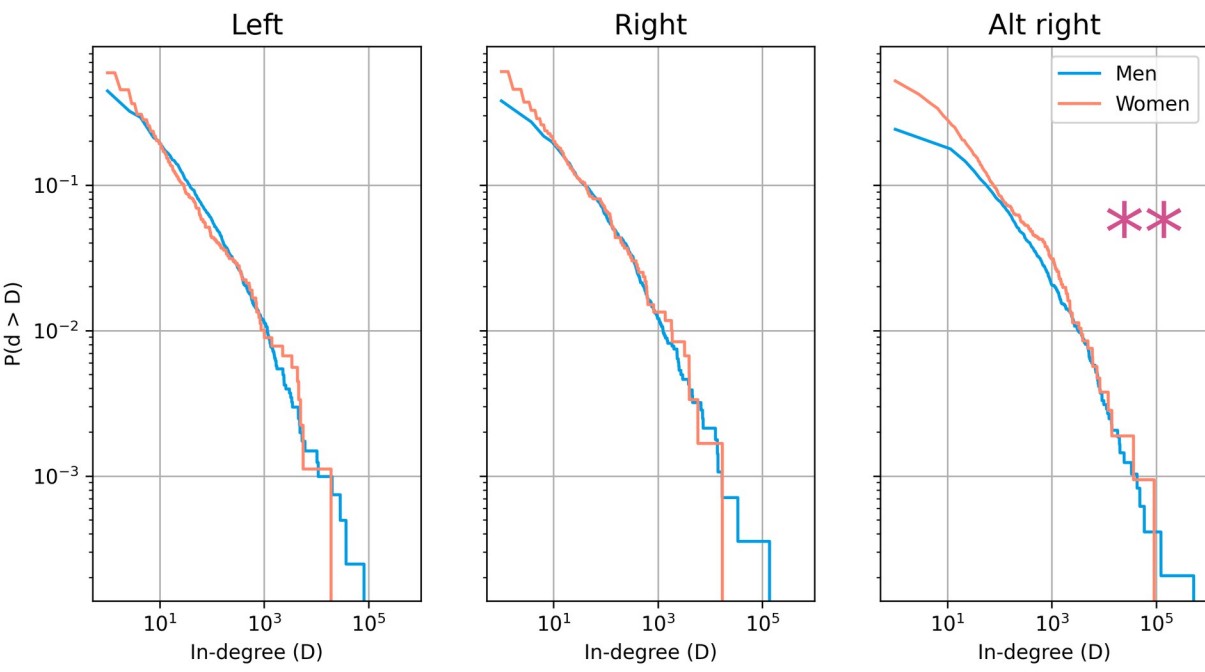

**Fig 2. The in-degree distributions of male and female politicians across the partisan-aligned subreddits.**

subreddits ($\mu = 57.3 \pm 81.6$) are significantly longer than those on the left ($\mu = 44.1 \pm 72.7$)($p < .0001; d = .17$) and alt-right ($\mu = 37.1 \pm 49.8$) ($p < .0001; d = .36$). Left-leaning comments are longer than those on the alt-right ($p < .0001; d = .13$). Post-hoc Tukey HSD tests found all interaction differences significant ($p < .0001$); they are visualized in Fig 3 and their effect sizes are reported in Table 2. Though there is a significant difference in comment length in all partisan groups, the effect size is only meaningful in the right-leaning subreddits, though it is small.

## 4.2 Combinatorial biases

*When female politicians are mentioned, are they mentioned in the context of other women? Or as a token women in a room of men?*

We find that women are most likely to appear in the context of other men than women, but men are also more likely to appear in the context of women than would be expected.

The observed values of $L(g_{given}, g_{add})$, as shown in Table 3 could, at first, be interpreted to suggest that male political entities are always more likely to be mentioned regardless whether one discusses male or female politicians. However, these values do not account for the relative number of male and female politicians being discussed.

The distribution of $L(g_{given}, g_{add})$ of null models of the dataset, as shown in Fig 4, shows that the observed values differ significantly from what is seen in random distributions of the data.

**Table 1. Kolmogorov-Smirnov test results comparing the distribution of comments per political entity across gender.**

|  | D | p-value |
|---|---|---|
| Left | .028 | >.05 |
| Right | .014 | >.05 |
| Alt-right | .057 | .007 |

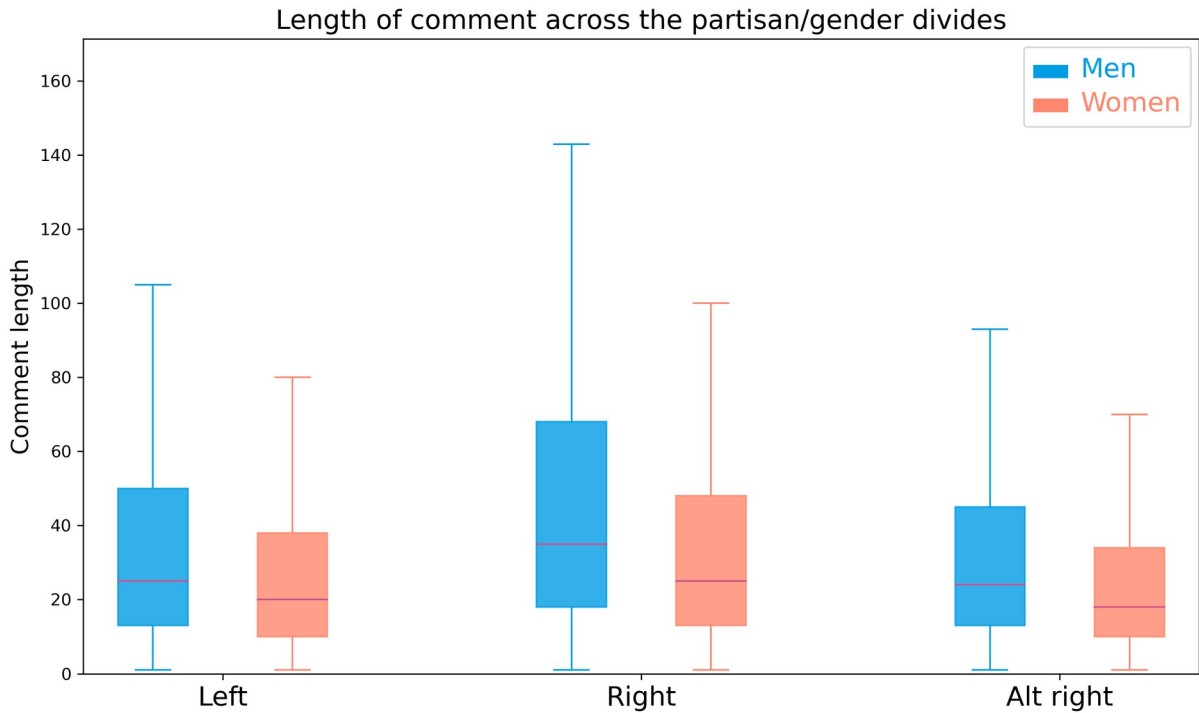

**Fig 3. The distribution of comment lengths according to gender and subreddit divide.**

Female politicians are significantly more likely to be mentioned when discussing either male or female politicians than would be expected from random permutations of the conversations. Male politicians are significantly more likely to be discussed in the context of female politicians and are significantly less likely to be discussed in the context of male politicians than would be expected from the null models. In all instances, the p value of the observed value is below $10^{-5}$.

**Table 2. Effect sizes of the comparisons visualised in Fig 3.** In bold are the within-partisan group comparisons. The larger value is the value on the left.

| Comparison | | Cohens d |
|---|---|---|
| Right, men | Alt-right, women | .44 |
| Right, men | Alt-right, men | .35 |
| Right, men | Left, women | .31 |
| Right, women | Alt-right, women | .30 |
| Left, men | Alt-right, women | .27 |
| **Right, men** | **Right, women** | **.20** |
| **Alt-right, men** | **Alt-right, women** | **.19** |
| Right, men | Left, men | .17 |
| **Left, men** | **Left, women** | **.16** |
| Right, women | Left, women | .14 |
| Left, men | Alt-right, men | .13 |
| Left, women | Alt-right, women | .12 |
| Right, women | Alt-right, men | .08 |
| Left, women | Alt-right, men | .08 |
| Left, men | Right, women | .04 |

**Table 3. Recorded values of $L(g_{given}, g_{add})$..**

|  |  | $g_{given}$ | |
|---|---|---|---|
|  |  | **female** | **male** |
| $g_{add}$ | **female** | 0.14 | 0.17 |
|  | **male** | 1.38 | 1.11 |

When looking at $L(g_{given}, g_{additional})$ values that share their $g_{additional}$ (and, therefore, their marginal probability), it appears that men are more likely to appear in the context of women than other men, and women are slightly more likely to appear in the context of men than other women. These results suggest gender heterophily in the dataset.

**4.2.1 Cross-partisan comparison.** We find, in all partisan divides, men are more likely to appear in the context of women. However, in the left- and right-leaning splits, women are more likely to be observed in the context of other women, rather than men. This is flipped in the case of alt-right subreddits.

As can be seen in Fig 5, all observed $L(g_{given}, g_{add})$ values differ significantly from those seen in the null distribution. In all instances, the p-value is less than $10^{-5}$. Therefore, there is a

## Recorded L(g_given, g_add) against null model distributions

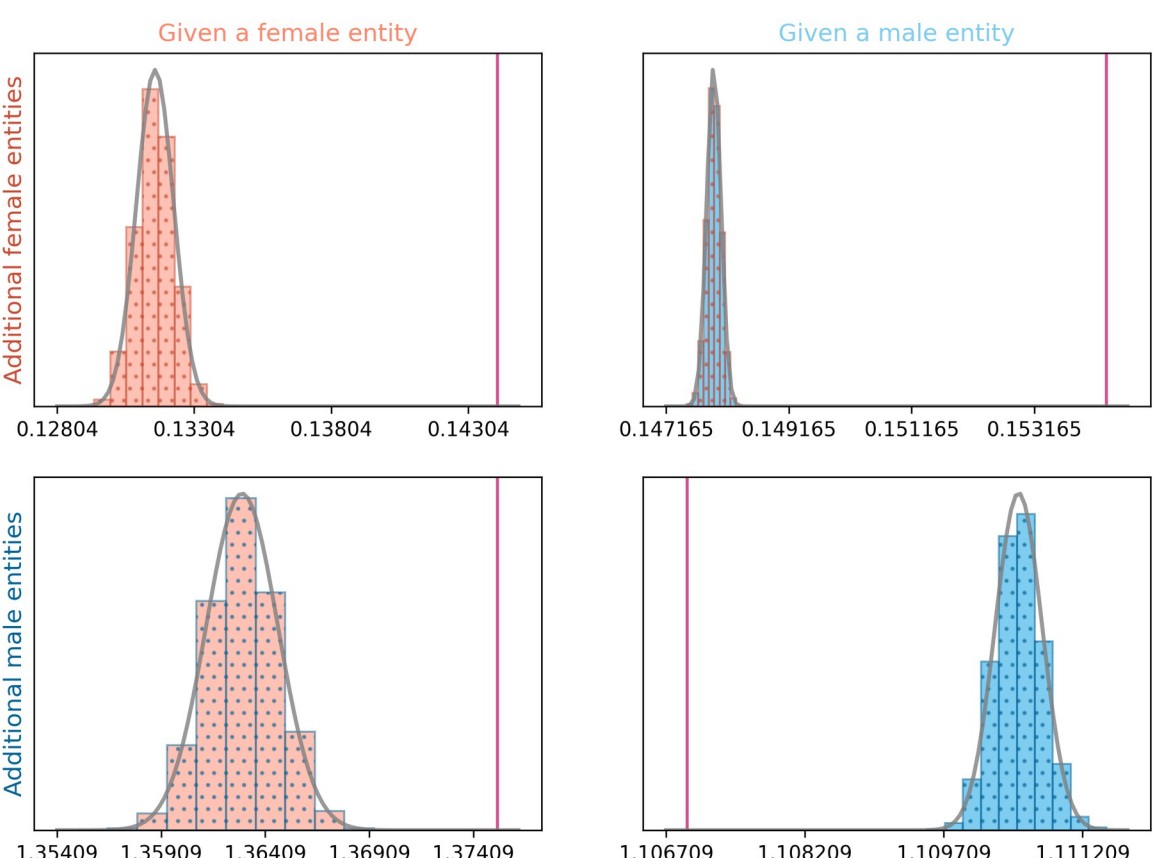

**Fig 4. The recorded value of $L(g_{given}, g_{add})$ is plotted in red against the recorded values from null models of the data.** A normal probability density function is fitted to the histogram.

Recorded L(g$_{given}$, g$_{add}$) against null model distributions

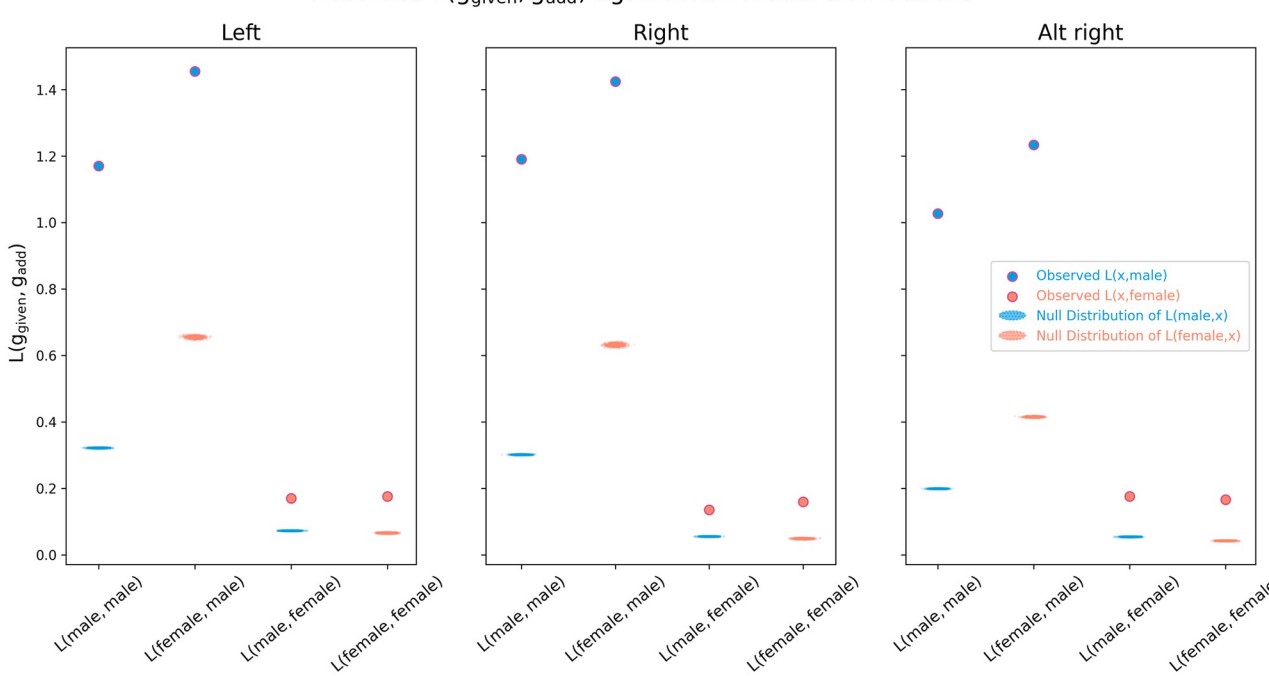

**Fig 5. The recorded values of $L(g_{given}, g_{add})$ plotted against the null distribution clouds of the left, right and alt-right leaning subreddits.** Given the large difference in values, we visualize the data as a marked observed value against a distribution cloud of expected values.

pattern in the combination of politicians discussed that cannot be approximated with random permutations of the genders. We report the obtained $L(g_{given}, g_{add})$ in Table 4. In all three partisan groups, $L(female, male)$ is greater than $L(male, male)$. $L(female, female)$ is greater than $L(male, female)$ in left and right-leaning subreddits. In the alt right subreddit, r/the_Donald, $L(male, female)$ is greater than $L(female, female)$.

### 4.3 Nominal biases

*Do people give equal respect in the names they use to refer to male and female politicians?*

We find that, while men are overwhelmingly named by their surname, women are much more likely to be named using their full name or given name.

A chi-square test of independence finds a significant relation between subject gender and referent used, $\chi^2(3, N = 13795685) = 2614058.47, p < .0001; V = .44$. The distribution of name choice across gender is pictured in Fig 6.

Male politicians are overwhelmingly named using their surname (69.68% of all instances), occasionally via their full name (16.90%), and rarely (2.12%) using their given name. In

**Table 4. Recorded values of $L(g_{given}, g_{add})$ on the cross-partisan dataset.**

|  |  | Left | | Right | | Alt-right | |
|---|---|---|---|---|---|---|---|
|  |  | $g_{given}$ | | $g_{given}$ | | $g_{given}$ | |
|  |  | male | female | male | female | male | female |
| $g_{add}$ | male | 1.17 | 1.54 | 1.19 | 1.42 | 1.03 | 1.23 |
|  | female | 0.17 | 0.18 | 0.14 | 0.16 | 0.18 | 0.17 |

## Name used in reference to politician

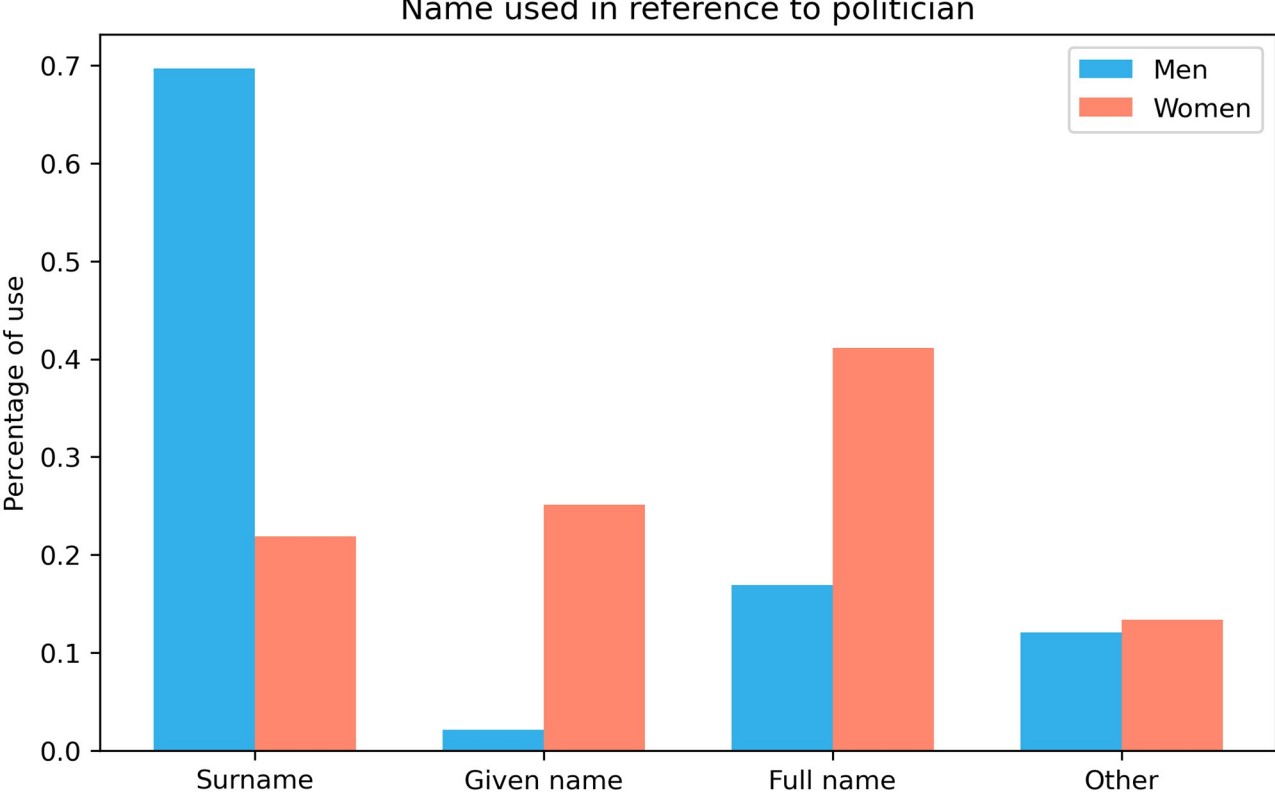

**Fig 6. The proportion of comments using the various names of an entity, across gender.** The "Other" category refers to any other names (i.e. pseudonyms, nicknames, misspellings).

contrast, women are most often named using their full name (41.14%), followed by their given name (25.09%) and surname (21.90%). Odds ratios indicate that the odds of a male politician being named by his surname are 8.14 times greater than for a female politician (95%$CI$: 8.12 − 8.17;$p$ <.0001). In contrast, the odds of a female politician being named by her first name are 15.24 times greater than for a man (95%$CI$: 15.2 − 15.3;$p$ <.0001). Women politicians have odds 3.38 times greater than men to be named using their full name (95%$CI$: 3.37 − 3.39;$p$ <.0001). Given the vague nature of the "other" category, we do not analyse that value further, but we do note that men and women are equally likely to be referred to by "other" names.

These results suggest that male politicians are more likely to be approached professionally. On the other hand, female politicians are significantly more often mentioned with their given names, which could originate from lacking respect towards them and, ultimately, gender bias.

**4.3.1 Cross-partisan comparison.** We find that women are much more likely to be named by their given name in alt-right and right-leaning subreddits than in left-leaning splits. However, in all partisan splits, men are significantly more likely to be named via their surname than women.

When looking across the partisan divide, a three-way loglinear analysis produces a final model retaining all effects with a likelihood ratio of $\chi^2(0) = 0$, $p = 1$, indicating that the highest-order interaction (partisan group x gender x name choice interaction) is significant ($\chi^2(6) = 3844.066$, $p$ <.0001). Further separate chi-square tests are then performed on two-way interactions. In left leaning subreddits, there is a significant association between politician gender and choice of nomination ($\chi^2(3) = 64204$, $p$ <.0001, $V = .39$); this holds true for right-leaning

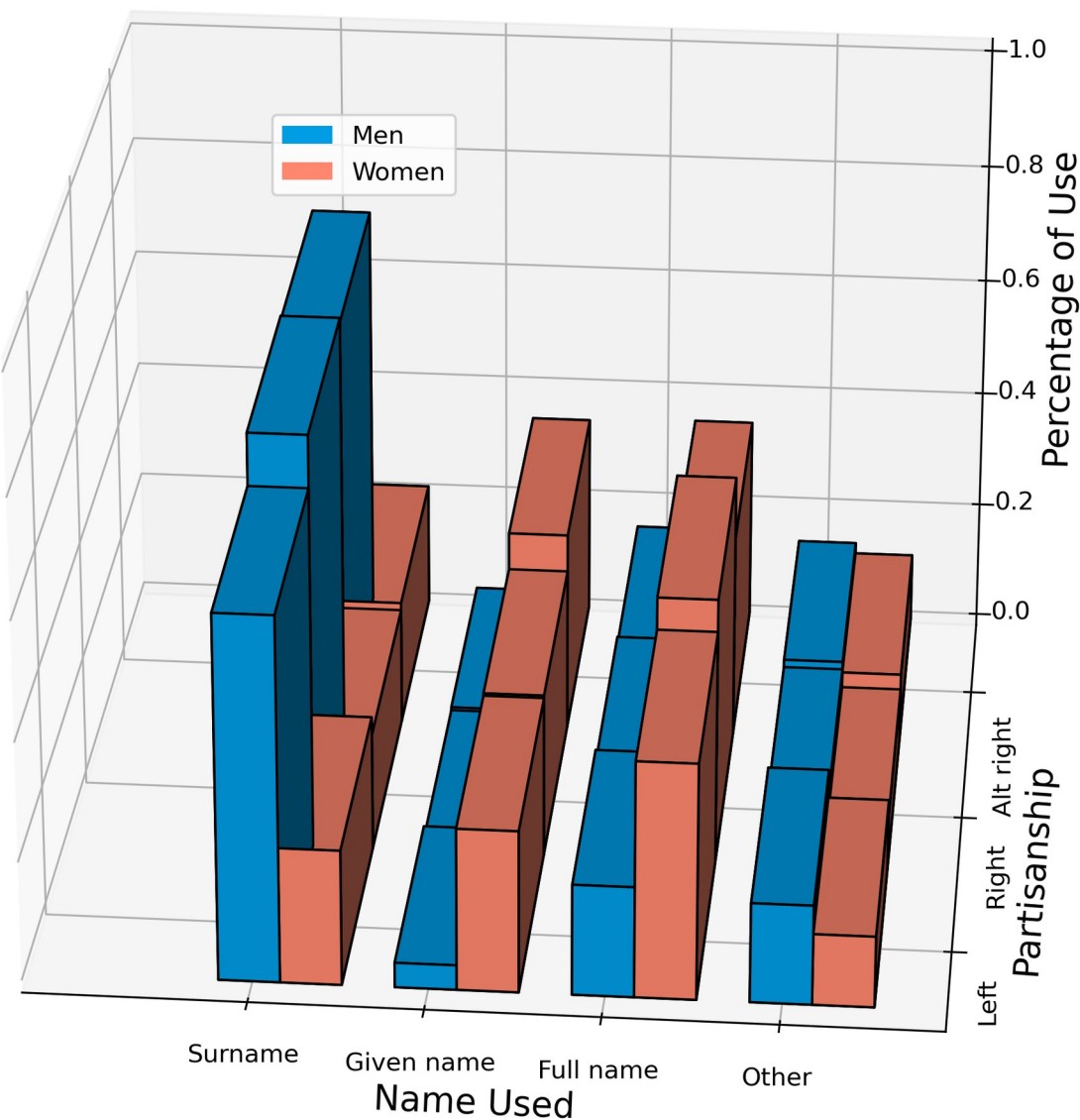

**Fig 7. An expansion of the use of nameage for politicians across the partisan divide of the data (see along y axis).**

subreddits ($\chi^2(3)$ = 93828, $p$ <.0001, $V$ = .47) and the alt-right subreddit ($\chi^2(3)$ = 416638, $p$ <.0001, $V$ = .50). There is also a significant association between partisan divide and choice of nomination for female politicians ($\chi^2(6)$ = 2874.6, $p$ <.0001; $V$ = .06), and male politicians ($\chi^2(6)$ = 11585, $p$ <.0001, $V$ = .05). In all three divides, Fig 7 shows a similar pattern; Men are named by their surname, but women are named by their full name or given name.

In alt-right subreddits, the odds ratio for a woman to be named by her given name is 18.8 times greater than for a man (95%$CI$: 18.5 − 19.0; $p$ <.0001). In right-leaning subreddits, the odds are 17.4 (95%$CI$: 16.9 − 17.9; $p$ <.0001). While the odds for women to be named by her first name are still 8.5 times greater than for men in left leaning subreddits (95%$CI$: 8.3 − 8.7; $p$ <.0001); these odds are half of those are seen in the right-leaning and alt-right subreddits. Collapsing surname use and full-name use as a "professional" reference of a politician, the odds of a woman politician being named in a "professional" manner is 1.31 greater in left-leaning

subreddits (95%$CI$: 1.29 − 1.33;$p$ <.0001) and 1.45 greater in right-leaning subreddits than in the alt-right subreddit, /r/the_Donald (95%$CI$: 1.42 − 1.48;$p$ <.0001). The difference between left and right-leaning subreddits is insignificant ($p$ >.01).

When it comes to men, odds-ratios show they are 9.5 times more likely to be named by their surname than women in right-leaning subreddits (95%$CI$: 9.3 − 9.8;$p$ <.0001); the odds are 8.6 (95%$CI$: 8.5 − 8.7;$p$ <.0001) in the alt-right subreddit, /r/the_Donald. This difference is nearly double the difference seen in left-leaning subreddit; In left-leaning subreddits, the odds for men to be named by their surname are just 5.4 times greater than women (95%$CI$: 5.3 − 5.5;$p$ <.0001).

## 4.4 Sentimental biases

*When people discuss male and female politicians, do they express equal sentiment and power levels in the words chosen?*

We find no large difference in sentiment and power attributed to male and female politicians.

Student t-tests of the lexicon-based measure show that comments about male politicians ($N$ = 7190149;$\mu$ = 0.325 ± 0.204) have greater valence than comments about female politicians ($N$ = 980553;$\mu$ = 0.314 ± 0.204) ($t$(8170700) = 47.78, $p$ <.0001;$d$ = 0.05). Comments about male politicians ($\mu$ = 0.300 ± 0.187) also show significantly greater dominance than those about women ($\mu$ = 0.285 ± 0.184) ($t$(8170700) = 74.31, $p$ <.0001;$d$ = 0.08). It should be noted that, though significant, the effect sizes (as measured via Cohen's D) of these differences are negligible (<0.2).

A chi-square test of independence finds a significant relation between subject gender and referent used, $\chi^2$(1, $N$ = 8170794) = 3811.0, $p$ <.0001, $V$ = .02. Odds ratio tests show that men are 1.17 more likely than women to be in a comment of positive sentiment (95%$CI$: 1.16 − 1.18;$p$ <.0001). Though this value is significant, the strength of the association (as measured via Cramer's V) is negligible.

Our results agree with previous studies that find female politicians to have lower dominance and sentiment attributed to them. However, we treat these results with a grain of salt, since the differences are only negligible.

**4.4.1 Cross-partisan analysis.** We find that most differences in sentiment and dominance across the partisan-splits are negligible. Men on alt-right subreddits have significantly higher sentiment and dominance than women on left and right-leaning subreddits.

The average comment valence and dominance scores show a bi-modal distribution in all subreddits, as can be seen in Figs 8 and 9. However, the sample size is sufficiently large to continue with parametric statistical tests.

A two-way ANOVA of average comment valence, as measured via the sentiment lexicon, finds a significant interaction of gender and partisanship ($F$(2, 1598999) = 89.67;$p$ <.0001) as well as significant main effects of gender ($F$(1, 1598999) = 775.13;$p$ <.0001) and group ($F$(2, 1598999) = 4299.79;$p$ <.0001). The data is shown in Fig 8 and Table 5. Post-hoc Tukey HSD tests show that comments about men ($N$ = 1363556;$\mu$ = .336 ±.209) have higher valence than comments about women($N$ = 235449;$\mu$ = .323 ±.207) ($p$ <.0001;$d$ = 0.05). Comments in left-leaning subreddits ($N$ = 234430;$\mu$ = .307 ±.202) have significantly lower valence than comments in right-leaning ($N$ = 242075;$\mu$ = .316 ±.202) ($p$ <.0001;$d$ = 0.04) and alt-right subreddits ($N$ = 1122500;$\mu$ = .344 ±.211) ($p$ <.0001;$d$ = 0.18). Comments in right-leaning subreddits have significantly lower valence than the alt-right subreddit ($p$ <.0001;$d$ = 0.13). The effect sizes of all differences are negligible. Post-hoc Tukey HSD tests of interactions find a significant difference ($p$ <.0001) for nearly all pairs, except ($p$ >.05) in the average valence of male

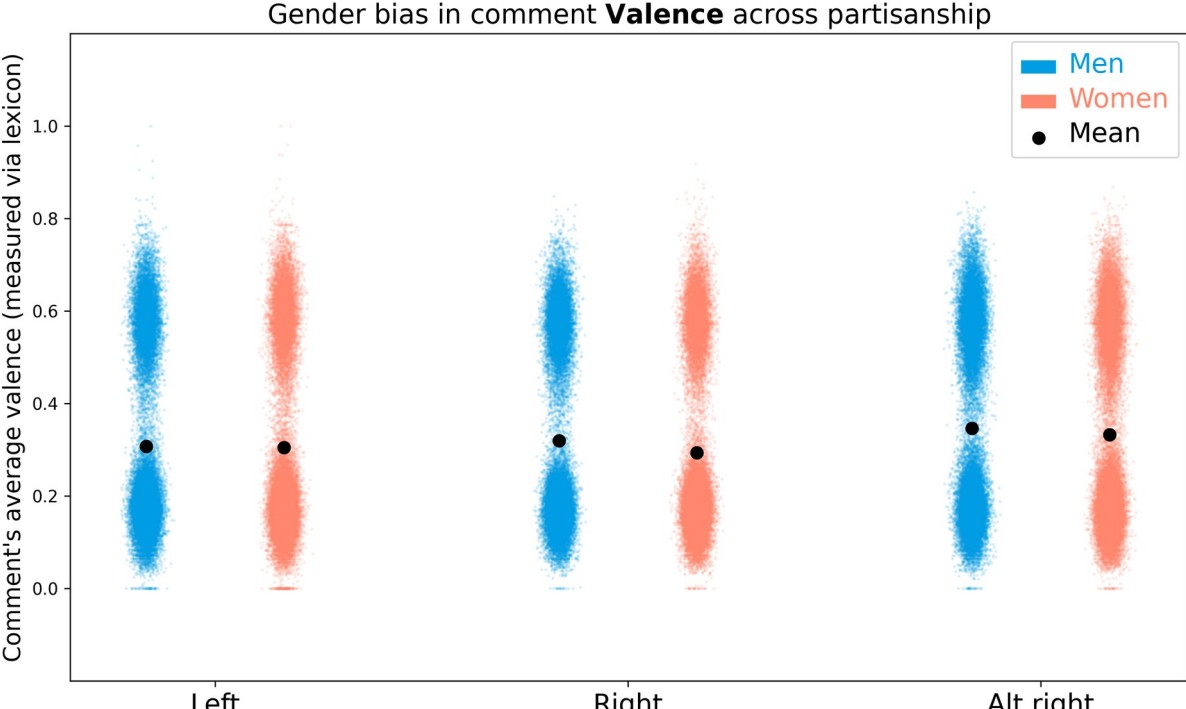

**Fig 8. Visualization of the distribution of average comment valence across subreddits and topic gender.**

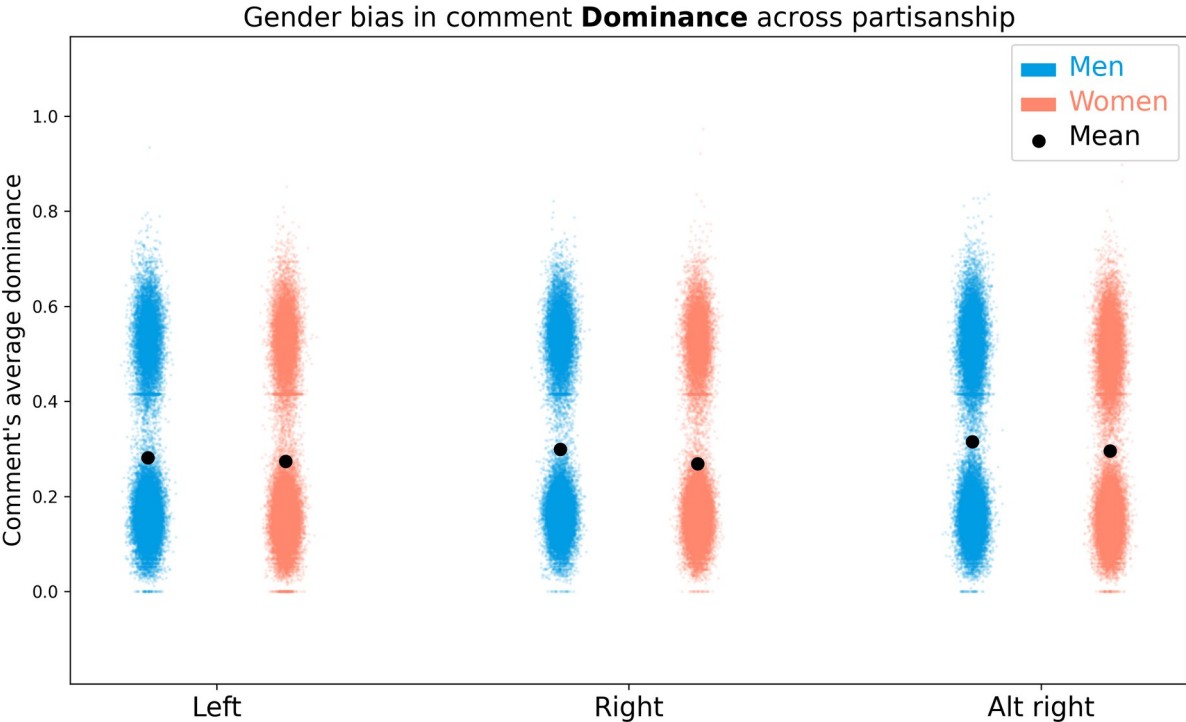

**Fig 9. Visualization of the distribution of average comment dominance across subreddits and topic gender.**

**Table 5. Results of Tukey-HSD tests on comment mean valance across partisanship and subject gender.** The non-negligible effect sizes are bolded.

| Comparison | | differences | p-value | Cohen's d |
|---|---|---|---|---|
| Alt right, men | Right, women | **0.05** | <.0001 | **0.25** |
| Alt right, men | Left, women | **0.04** | <.0001 | **0.2** |
| Alt right, men | Left, men | **0.04** | <.0001 | **0.19** |
| Alt right, women | Right, women | **0.04** | <.0001 | **0.19** |
| Alt right, women | Left, women | **0.03** | <.0001 | **0.13** |
| Alt right, men | Right, men | **0.03** | <.0001 | **0.13** |
| Right, men | Right, women | **0.03** | <.0001 | **0.12** |
| Alt right, women | Left, men | **0.03** | <.0001 | **0.12** |
| Left, women | Right, women | **0.03** | <.0001 | **0.06** |
| Right, men | Left, women | **0.01** | <.0001 | **0.07** |
| Alt right, men | Alt right, women | **0.01** | <.0001 | **0.07** |
| Left, men | Right, women | **0.01** | <.0001 | **0.06** |
| Right, men | Left, men | **0.01** | <.0001 | **0.06** |
| Alt right, women | Right, men | **0.01** | <.0001 | **0.06** |
| Left, women | Right, women | **0.01** | <.0001 | **0.07** |
| Left, women | Left, men | — | n.s. | — |

and female politicians on left-leaning subreddits. However, many of these effect sizes are negligible.

A two-way ANOVA of average comment dominance finds a significant interaction of gender and partisanship ($F(2, 1598999) = 103.5; p < .0001$) as well as significant main effects of gender ($F(1, 1598999) = 1960.6; p < .0001$) and group ($F(2, 1598999) = 3238.5; p < .0001$). The data is shown in Fig 9 and Table 6. Post-hoc Tukey HSD tests show that comments about men ($\mu = .258 \pm .160$) have higher dominance than comments about women($\mu = .244 \pm .157$) ($p < .0001; d = 0.10$). Comments in left-leaning subreddits ($\mu = .280 \pm .184$) have higher significantly lower dominance than comments in right-leaning ($\mu = .295 \pm .187$) ($p < .0001; d = 0.08$) and alt-right subreddits ($\mu = .312 \pm .190$) ($p < .0001; d = 0.17$). Comments in right-leaning subreddits have significantly lower dominance than the alt-right subreddit ($p < .0001; d = 0.09$).

**Table 6. Results of Tukey-HSD tests on comment average dominance across partisanship and subject gender.** The non-negligible effect sizes are bolded.

| Comparison | | differences | p-value | Cohen's d |
|---|---|---|---|---|
| Alt right, men | Right,women | 0.05 | <.0001 | **0.25** |
| Alt right, men | Left,women | 0.04 | <.0001 | **0.22** |
| Alt right, men | Left,men | 0.03 | <.0001 | 0.18 |
| Right, men | Right,women | 0.03 | <.0001 | 0.16 |
| Alt right, women | Right,women | 0.03 | <.0001 | 0.15 |
| Right, men | Left,women | 0.02 | <.0001 | 0.13 |
| Alt right, women | Left,women | 0.02 | <.0001 | 0.12 |
| Alt right, men | Alt right, women | 0.02 | <.0001 | 0.1 |
| Right, men | Left,men | 0.02 | <.0001 | 0.09 |
| Alt right, men | Right, men | 0.02 | <.0001 | 0.09 |
| Alt right, women | Left,men | 0.01 | <.0001 | 0.08 |
| Left,men | Right,women | 0.01 | <.0001 | 0.07 |
| Left,men | Left,women | 0.01 | <.0001 | 0.04 |
| Right,women | Left,women | 0.01 | 0.002 | 0.03 |
| Right, men | Alt right, women | 0.003 | <.0001 | 0.02 |

Post-hoc Tukey HSD tests of interactions find significant difference ($p < .01$) for all pairs, though many of the effect sizes are negligible.

A three-way loglinear analysis of the classifier-output sentiment labels produces a final model retaining all effects with a likelihood ratio of $\chi^2(0) = 0$, $p = 1$, indicating that the highest-order interaction (partisan group x gender x sentiment label) is significant ($\chi^2(7) = 9141.02$, $p < .0001$). Further separate chi-square tests are then performed on two-way interactions. In left leaning subreddits, there is a significant association between politician gender and comment sentiment ($\chi^2(1) = 106.96$, $p < .0001$, $V = .02$); this holds true for the alt-right subreddit ($\chi^2(1) = 2194.3$, $p < .0001$, $V = .04$), but not right-leaning subreddits ($\chi^2(1) = 2.2$, $p > .05$). There is also a significant association between partisan divide and comment sentiment label for female politicians ($\chi^2(2) = 1039.2$, $p < .0001; V = .07$), and male politicians ($\chi^2(2) = 5032.8$, $p < .0001$, $V = .06$).

In left leaning subreddits, men are 0.87 times as likely as women to be named in a comment with positive sentiment (95%$CI$: $0.85 - 0.90; p < .0001$). In contrast, in the alt-right subreddit, men are 1.34 times more likely to be described in positive sentiment than women (95%$CI$: $1.33 - 1.36; p < .0001$) However, while we see significant differences, the strength of these associations, as measured via Cramer's V, are minimal. The frequencies are visualized in Fig 10.

## 4.5 Lexical biases

*When people discuss male and female politicians, how do the words they use to describe them differ?*

We find highly female-biased words are more likely to be about body, clothing or family-related descriptors than male-biased words. In contrast, highly male-biased words are more likely to be profession-related.

Firstly, we investigate the differences in results between the traditional and improved PMI method, to show the efficacy of the entity-based approach. Thereafter, we go into the general gender biases that can been seen in the dataset. Again, we conclude with the cross-partisan analysis. Given the source of this dataset, there may be some offensive and/or explicit words in the following sections and associated appendices.

The top 10 female-associated words obtained through traditional and entity-based PMI methods are shown in Table 7. While also including explicitly gendered words (e.g. 'chairwoman'), the traditional PMI method attributes a high PMI value to many ethnicity-centered words, such as 'Somalian' and 'Cherokee'. These should not be gendered words. They are, however, heavily associated with specific popular politicians, Somalian-born US Representative Ilhan Omar and Senator Elizabeth Warren, who has faced criticism about claims of Cherokee-heritage.

In contrast, the entity-based approach removes many of these obviously confounding words related to ethnicity. In addition, many obviously gendered words are prioritized (e.g. "chairwoman" and 'menopausal'). Though no longer on the top-10 list, 'spokeswoman' still has a high female PMIe (1.83, #17 on the list). Less-obviously gendered terms are retained in both lists (e.g. 'pantsuit'), but more appear in the PMIe word list. These terms are further explored in the next sections.

**4.5.1 General gender comparison.**  Using this entity-based PMI approach, we now turn to the words in the dataset that have a high gendered PMI value. The top 10 for each explored gender with their PMI value for that gender are shown in Table 8. The remainder of the words, and their annotated senses, are visible in S2 Table in §6. It is interesting to note that the highest PMI values for the male-biased words are quite low and near zero; this may be due to the overwhelming existence of more male entities in the dataset. However, despite this near-zero

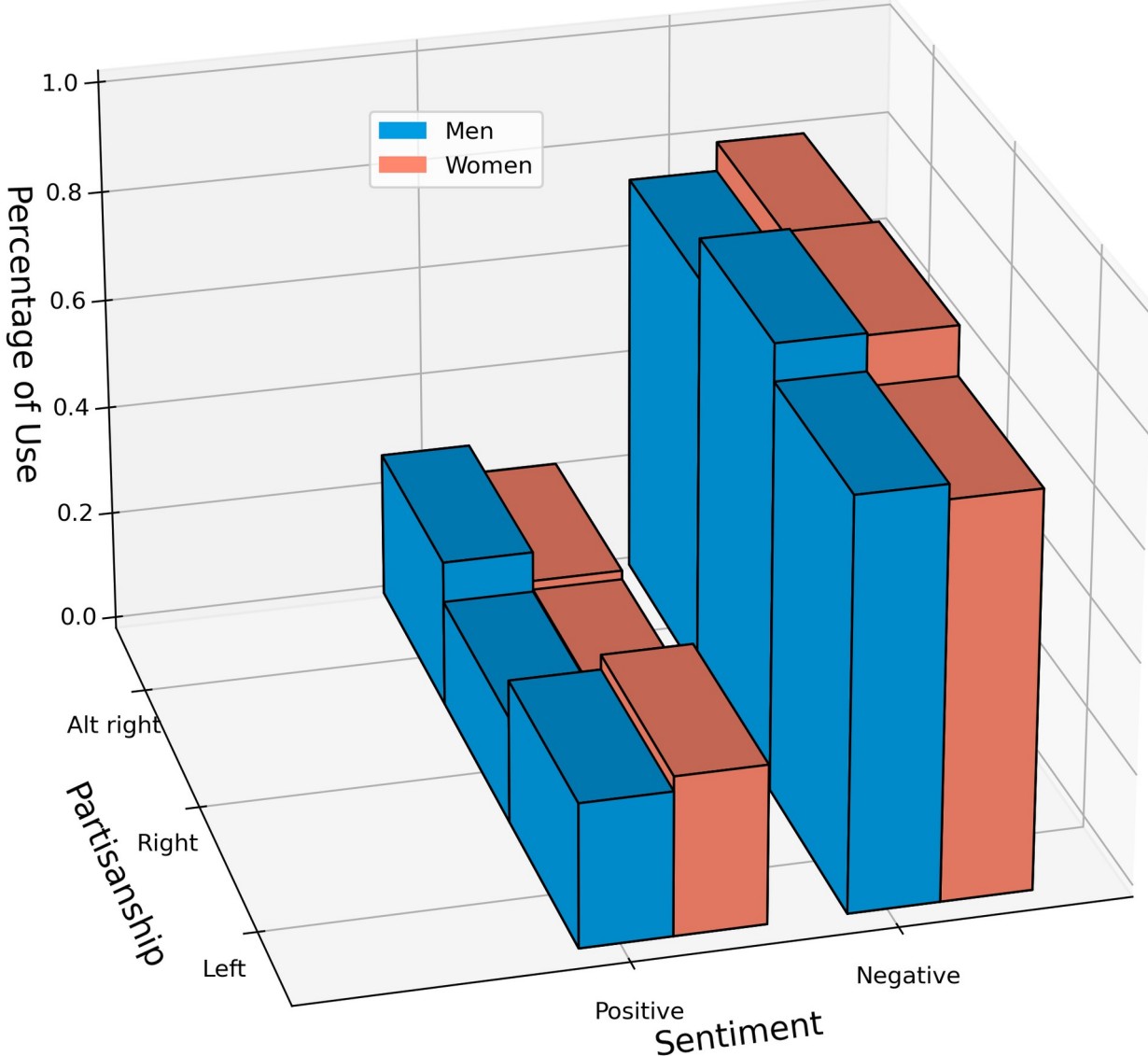

**Fig 10. The output of the sentiment classifier is compared across gender and subreddit partisanship.** Though gender differences are significant in both left-leaning and alt-right subreddits, the assocation strength is negligible, like the differences measured via the sentiment lexicon.

number, there are some obviously gendered words (e.g. 'prince') on the list. An ethnicity-related term remains the male-biased PMIe list ('turkish'). However, this may owe to actual gender bias, given Turkey's low ranking on the gender gap report [4], especially within political empowerment. Within the top-100 list of male and female-biased words, all PMI-values are above 0.10.

Though we hoped to use lexica to analyse the larger dataset, the Slang SD, NRC, and Supersenses lexica together only cover 32.5% of the top 100 words for either male or female bias. Therefore, we rely on hand-coded senses and sentiments. The annotators achieve a Cohen's Kappa Agreement of 0.618 on the Handcoded Sentiment and 0.620 on Senses on a subsample of 50 words, suggesting substantial agreement.

**Table 7. Results of the traditional PMI on the left.** The entity-based technique is on the right. See the elimination of ethnicity-based terms on the right, while some words like "pantsuit" are still retained in the top 10.

| Traditional PMI | | PMIe | |
|---|---|---|---|
| truancy | 1.969493 | chairwoman | 1.421482 |
| react | 1.936892 | pantsuit | 1.407579 |
| somalian | 1.93603 | matriarch | 1.351489 |
| cherokee | 1.927183 | facelift | 1.313268 |
| cheekbone | 1.92447 | menopausal | 1.313268 |
| pantsuit | 1.901174 | harpy | 1.313268 |
| beetus | 1.893833 | scarf | 1.29525 |
| directory | 1.888102 | clitoris | 1.264478 |
| spokeswoman | 1.874243 | clit | 1.264478 |
| jamaican | 1.866066 | brunette | 1.264478 |

A chi-square test of independence is performed to examine the relation between gender and the distribution of word senses. The relation between these variables is significant ($\chi^2(7, N = 200) = 46.29, p < .0001; V = .50$) and their distributions are visualized in Fig 11. Overall, there are few positive words in both the top male or female-biased words. Negative labels make up a high portion of both male and female-skewed words; Odds ratio tests show that the odds of finding negative labels in male-biased words are not significantly higher than in female-biased words ($p > .05$). In addition, the odds of female-biased words containing attribute-related descriptors (18%) are not significantly greater than male-biased words (13%) ($p > .05$). However, there is a big disparity between the genders in the remaining categories. The odds of male-skewed words containing profession and belief-related descriptors (20%) are 2.98 greater than female-skewed words (7%) ($95\% CI: 1.16 - 7.22; p < .05$). In contrast, the odds of female-skewed words containing body-related descriptors (19%) are an estimated 8.94 times greater than for male-skewed words (3%) ($95\% CI: 2.5 - 31.4; p < .0001$). In addition, while there are 0 male-biased words related to their clothing and family, both categories are represented more in female-biased words (7% and 6%) than words related to their own profession (4%).

Hence, words associated with female politicians are often irrelevant descriptors of their appearance or family. This treatment does not apply to male politicians who are described in relation to their profession and politics in general. These results suggest presence of gender bias on a lexical level.

**Table 8. The top-10 male and female-biased words in the dataset, using the entity-based PMI approach.**

| Male-bias | | Female-bias | |
|---|---|---|---|
| bloke | 0.171301 | chairwoman | 1.421482 |
| wanker | 0.166013 | pantsuit | 1.407579 |
| prince | 0.160944 | matriarch | 1.351489 |
| lawful | 0.159782 | facelift | 1.313268 |
| turkish | 0.151414 | menopausal | 1.313268 |
| madman | 0.147948 | harpy | 1.313268 |
| unchecked | 0.146697 | scarf | 1.29525 |
| punchable | 0.145394 | clitoris | 1.264478 |
| dickhead | 0.142614 | clit | 1.264478 |
| truck | 0.142614 | brunette | 1.264478 |

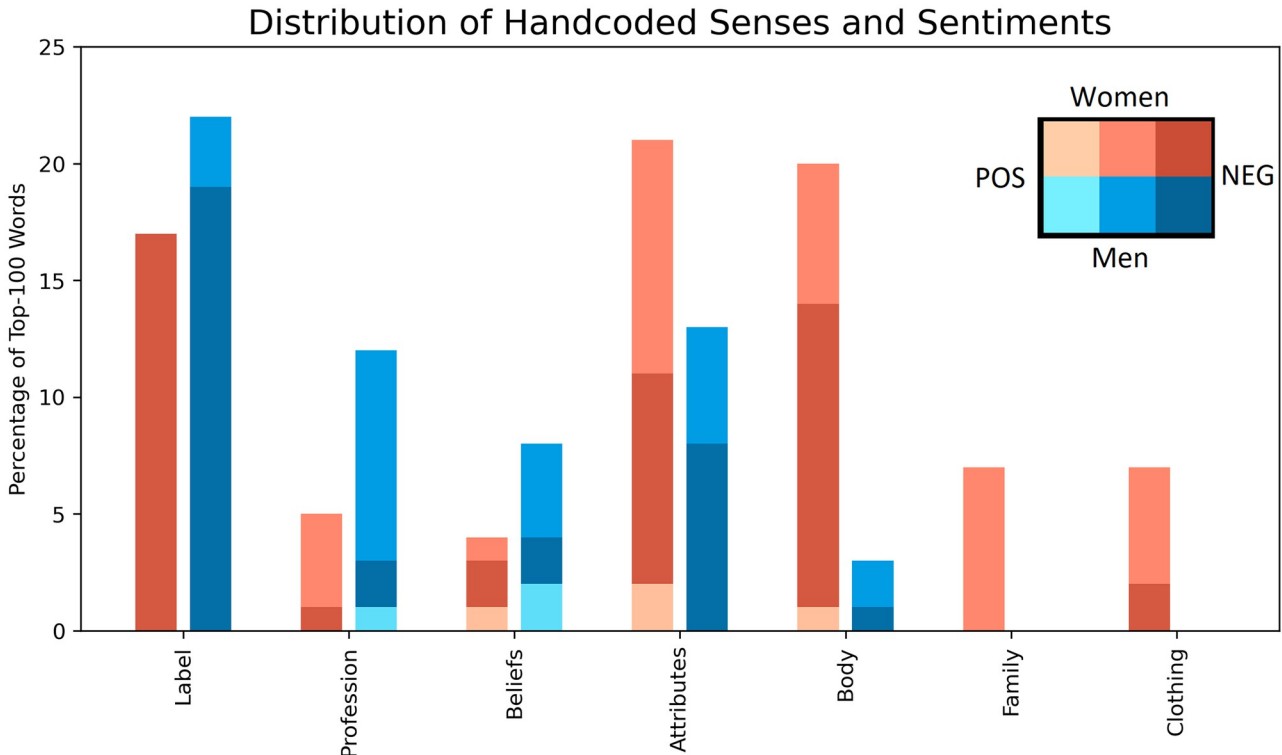

**Fig 11. The distribution of the handcoded senses and sentiments across words with high gender bias.** Words coded as "Other" are not included.

**4.5.2 Cross-partisan comparison.** We find the alt-right subreddits contain much more body-related descriptors for female politicians than left or right-leaning subreddits.

When comparing the hand-coded senses and sentiments of the top gendered words in the partisan-divided groups of subreddits, as shown in Fig 12, there is a visual difference in which

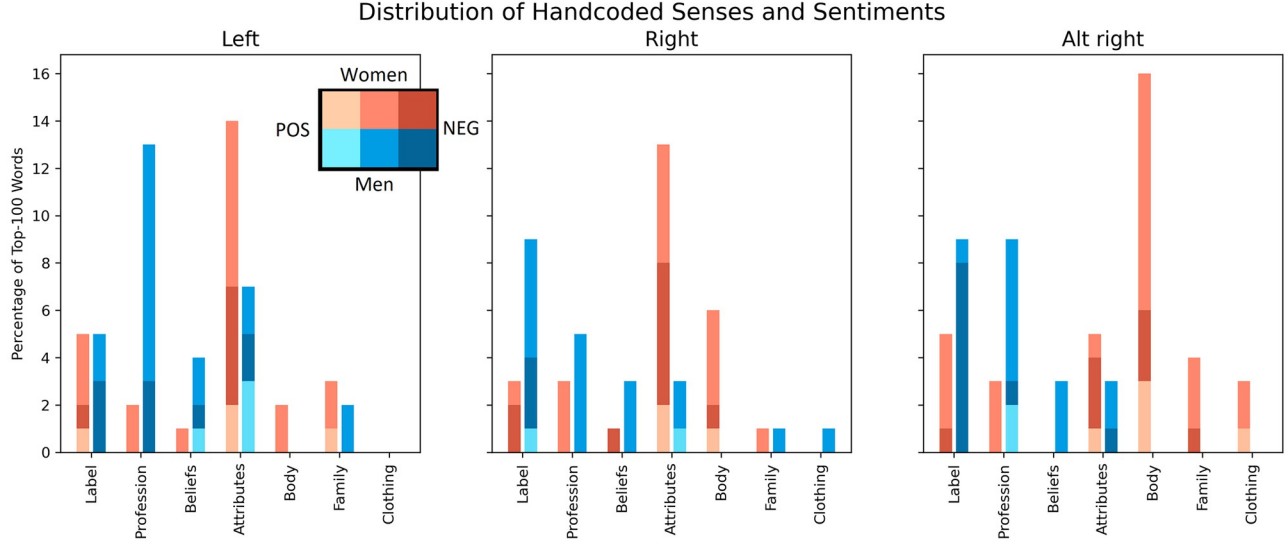

**Fig 12. The word sense distributions across the most gender biased words along the partisan-leaning subreddits.**

descriptors play larger roles across the genders. While body-related descriptors appear for women in all three groups, they do not appear in either of the three male-skewed lists. However, odds ratio show that body-related descriptors have 8.00 times greater odds of being highly female-biased on the alt-right subreddit than on left-leaning subreddits (95%$CI$: 1.75 − 36.6;$p$ <.01). There are no other significant differences in body-related descriptors between subreddit groups ($p$ >.05). Instead, on the left and right-leaning subreddits, women are more described by their attributes. The odds of woman-related words being attribute-related are 3.49 times greater on the left-leaning and right-leaning subreddits than on the alt-right subreddit (95%$CI$: 1.15 − 10.63;$p$ <.05). There is no difference between the left and right-leaning subreddits.

On the other hand, in left-leaning subreddits, male-skewed words have 8.07 greater odds of containing profession- and belief-related words than female-skewed words (95%$CI$: 2.19 − 29.8;$p$ <.001). The odds in the alt-right subreddits are almost half of that: 4.95 (95%:1.30 − 18.8;$p$ <.05). The odds are not significantly different in right-leaning subreddits ($p$ >.05). However, looking at the distribution, it appears that these differences are due to fewer profession-related words appearing on the male-skewed list, not more female-biased profession-related words.

## 5 Discussion

### 5.1 Biases in the general community

In our investigation of coverage biases (§4.1), we find a generally equal amount of public interest in both male and female politicians; male and female politicians generate equivalent distributions of comments (Fig 1). Furthermore, the differences in the proportion of possible politicians discussed (a slightly greater proportion of female politicians are discussed) and comment length (comments about female politicians are, on average, 6 tokens shorter than those for male politicians) are negligible. While previous studies have shown longer articles and greater coverage devoted to male figures [25, 26], especially male politicians [43], these are not measures of the public's interest, but that of the media's (which may be affected by the intended audience, sponsors, and an editorial hierarchy). In contrast, our measures of coverage are measures of general public interest. This is interesting, given that another study of public interest, as measured via Wikipedia article views, suggests that, though interest is generally higher for female figures than for male figures, this is not the case for politicians [43]. We find equal public interest, likely as we measure a higher standard of engaged interest. However, our comparisons of coverage and public interest are done quite simply, without consideration of the politician's age and level of position. These two factors are likely to affect the amount of attained public interest (and, due to known gender biases, are most likely to act against female politicians, as they are more likely to have shorter careers and stay in lower levels of hierarchy [73–75]). Therefore, there remains the possibility that our finding of equal public interest in politicians is a conservative estimate, and there may be even greater public interest in female politicians to equivalent male politicians.

Interestingly, the null models conducted in this investigation of combinatorial biases (§4.2) suggest that female politicians are more likely to be mentioned in the context of both other women and men than would be expected by their presence in the dataset (Fig 4). Though one may expect to see the Smurfette principle in practice [46], or, at least, gender homophily, it is surprising to see that women appear more often than would be expected in both men and women-containing conversations, and they are discussed in a manner that cannot be simulated with random permutations. Interestingly, when comparing values with shared marginal probabilities, it appears that men are more likely to appear in the context of women than other

men, and women are more likely to appear in the context of other men. If anything, this suggests heterophily within the network.

Starker biases begin to appear when one looks within the text rather than the overall structure, as we see in nominal (§4.3), sentimental (§4.4) and lexical (§4.5) analyses. Male politicians are more likely to be named professionally than female politicians. Instead, women are overwhelmingly more likely to be named using their given name than men (Fig 6). This validates many claims about female professionals being referred to using familiar terms, diminishing their authority and perceived credibility and widening the existent gender gap [49, 50]. Overall, female politicians are still most commonly referred to by their full name. However, it is important to note that the co-reference resolution step biases towards extending the longest observed name down the entire cascade. Therefore, the use of a full name may be over represented in this dataset. In addition, the named entity linker may miss many references to politicians under unknown nicknames or common first names, thereby excluding these comments from these analyses.

While female politicians have lower sentiment and dominance attributed to them in comments, the effect sizes are not meaningful. This is also seen when sentiment is measured via a classifier output. This is interesting, given that previous studies have shown that women generally have more positive sentiment and low dominance [22, 33, 53] attributed to them, a concept referred as benevolent sexism [17]. It could also be expected, from previous studies, that more negative sentiment would be expressed to women, given persisting implicit prejudices against female authority figures [5]. However, in our examination of sentiment, we do not see the manifestation of neither hostile nor benevolent sexism at play. It is interesting to note that, in the lexicon-based method, both the valence and dominance level comment averages show a bi-modal distribution– other studies using this lexicon show a normal distribution [76], which leads us to wonder from where the bimodality arises. Possibly, the two peaks belong to opposing party members (for example, the more positive peak corresponds to politicians of the same political alignment as the poster).

When it comes to the PMI-based lexical bias investigation, there is a clear difference in the most male and female-gendered words (Tables 7 and 8). Surprisingly, neutral and negative labels are equally likely to be heavily attributed to men or women. This echoes our investigation into sentimental biases; we do not see evidence for benevolence or hostility towards female politicians. However, there are still stark differences in how men and women are described. Profession and political belief-related terms show a heavy male-skew, echoing results of other studies [20, 24, 28]. This is not necessarily the case for women. Highly female-gendered words are often about irrelevant descriptors: their body, their clothing, and their family. This matches many gender bias studies which show that the public and media take a more personal interest into female professionals [20, 23, 28, 31, 33, 64, 65]. These results also match similar studies showing an overwhelming amount of body-related descriptors being attributed to women [23, 24, 27, 31]. However, while other studies show more positive body-related descriptors for women [31], we find predominately negative or neutral body-related descriptors. Though attribute-related descriptors appear in both male- and female-biased word lists, they are the only professional standards to which women are held. Not their policies or professional qualifications, but their other attributes: their elegance and their bossiness, if not their looks. The biases faced by female politicians lay outside benevolent or hostile sexism; they involve more nuanced societal expectations around their appearance and their personality. Finally, it is interesting to note that, Table 8 shows that even clearly male-biased words (e.g. "prince") have significantly lower PMI-values than female-biased words. While this likely owes to the predominance of male-centric comments in the dataset, it echos the recurrent theme in technology that men are the "null" or standard gender [77]. This is an unintentional

outcome from training algorithms on an imbalanced dataset, as one runs the risks of perpetuating existing biases.

## 5.2 Biases across the political spectrum

The sub-community nature of the dataset allows us to investigate how these observed biases change along the partisan line. In the left-leaning subreddits, there is what could be interpreted as the most egalitarian treatment across the genders, which coincide with expressed left-leaning values. These subreddits showed the most equal coverage of male and female politicians, in terms of the politicians mentioned, politician in-degree distribution, and comment lengths (§4.1.1). The odds of a female politician being named using her given name are half those seen in the right-leaning partisan divides (Fig 7). In addition, the left-leaning subreddits are the only subset that do not show a significant difference in sentiment between male and female politicians, though the difference observed in other interest groups is negligible (4.4.1). Compared to the two right-leaning divides, body-related descriptors are the least represented in the heavily female-biased words. However, some gender disparity still remains; male-skewed terms on left-leaning subreddit are overwhelmingly related to their profession and political beliefs, unlike heavily female-biased words (Fig 12). Men are held to a certain set of professional standards, whereas women politicians are described by their general attributes. Therefore, while left-leaning posters may discuss non-superficial attributes in female politicians, these attributes may not necessarily be politically relevant, but may showcase a different standard of qualifications that women are expected to uphold (e.g. trustworthiness, capability, likeability) instead of the professional qualities which which male politicians are described [78].

When it comes to the right-leaning subreddits, there are some conflicting results. The distribution of activity generated per entity is equal between men and women, though there are fewer comments about women, and fewer female politicians mentioned (§4.1.1). This group of subreddits show the greatest divide in the number of female and male politicians mentioned. In addition, comments about these female politicians are, on average, 10 tokens shorter, though the effect size is small. Taken together, it appears that participants in right-leaning subreddits have, overall, slightly less active interest in female politicians. Despite this lower engagement, however, female politicians are still treated with respect; they are equally as likely to be referenced in professional terms as in left-leaning subreddits, though women are twice as likely to be referenced by their given name than in those subreddits (Fig 7). Attribute-related descriptors make up a bigger proportion of heavily female-biased descriptors than body-related ones, and both profession and political-belief related descriptors are equally as likely to be female- and male-biased, unlike as seen the other political divides (Fig 12). However, this appears to arise from fewer male-biased profession-related descriptors, not more female-biased ones. Ultimately, in right-wing fora, the engagement in conversation about female politicians, though potentially smaller, remains professionally-relevant.

Finally, in the informal alternative-right subreddit, /r/the_Donald, there are much starker differences. Though many female politicians are mentioned, there is a significant difference in the level of interest generated by the politicians (Fig 2). Unlike the other political divides, combinatorial investigations suggest that women appear to be slightly more likely to be discussed in conjugation with men, rather than other women (Fig 5). When mentioned, female politicians are twice as likely to be named via their given name than the left-leaning subreddits, and they are almost 1.5 times less likely to be named using their full or surname than in both other political subreddit groups (Fig 7). The words most attributed to female politicians are overwhelmingly related to their bodies, rather than their profession, beliefs or other supraphysical attributes (Fig 12). This suggests an overall disregard for female politicians; not only is there

less active interest in the politicians, but, when they are discussed, female politicians are not as often discussed with respect as professionals but rather in relation to their body.

## 5.3 Limitations and contributions

There are some limitations in how far the cross-partisan analyses can be interpreted. Firstly, the obtained results cannot differentiate whether the observed differences are general patterns in the behaviour of the participants or differences in the actual politicians being discussed in the subreddits. Perhaps female politicians that prefer the use of their given name and have legitimate, professional reason to be described with body-related terms (e.g. disease awareness activists) are more likely to be discussed on the alt-right. Our grouping of these subreddits may also affect the observed results. Even within similar communities, gender biases and community norms may differ, creating a noisy sample [35]. In addition, the population of the left-leaning and right-leaning subreddits may not be equally disparate as those of the right-leaning and alt-right subreddits. Given the overlap in political belief, the population of posters may also overlap. The observed differences in this study may be generated by political viewpoints or other differences between the subreddits (e.g. moderation level or formality). These analyses simply showcase the language that is acceptable within the community, after moderation, which may differ both across community formality and partisanship. Other studies simply investigate the right-left divide [28]. To ensure that the overwhelming presence of Trump-related comments in the dataset did not results, in S1 Text in §6, we validate that we continue to see similar patterns of results even with the removal of all Trump-related comments from the data-set. Therefore, the biases we describe are not necessarily simply differences between women and Donald Trump but gender differences that can be applied more generally across politicians. We continue to see the same pattern of results even in the alt-right data, which is heavily influenced by Donald Trump. Therefore, we would like to stress that these observed biases are not necessarily guided by certain prominent figures but are reflective of biases within the general ideologies.

While Reddit users make up a wider variation of people than news journalists, Wikipedia editors, and book authors [20, 24, 31, 43, 79], the population from which many other studies draw their data, the Reddit user distribution is still skewed towards white, college-educated men [40], though we take efforts to increase the dataset's diversity. In addition, our entity-linker showed only 50% accuracy when linking to female politicians, giving us fewer comments about female politicians. This is a clear gender bias in the data-processing step; We suspect that, given that female politicians appear relatively likely to be referenced by their given name, surname or full name, this may be a source of noise for the entity-linker that contributes to its inaccuracy. More investigation on how gender bias emerges in these intermediate processing steps is warranted, as it likely contributes to some skew within the dataset. Other processing steps may contain gender biases or may affect the measured biases; the coreference resolution pre-processing step likely biases our dataset towards more instances of full names in the comment text. Many NLP tools are trained on formal text (i.e. books, newspapers) and may not be as effective on social media text, like that seen in Reddit. To avoid these issues, we choose simpler pre-processing steps and avoid the use of parsers, but errors still arise from the pre-processing that is conducted, given the text medium.

Due to the limitation in the multilingual pre-processing tools required for this investigation, we focused our investigation in English. However, other studies have found that both linguistic and extra-linguistic biases can vary heavily across language [20]. Therefore, we cannot necessarily generalise our findings across languages and non-English-speaking cultures. While one could translate all text into a single language for analysis, this runs the risk of amplifying

biases present in machine (or human) translators, rather than in the source language. Similar yet multilingual studies would benefit from multilingual entity linker and coreference resolution tools to create the required data-set. Most of described analyses should be feasible in a variety of other languages. The spaCy dependency parser used to determine descriptors for the lexical bias assessment is available in 21 different languages. However, both the lexicon and classifier-based methods for sentimental bias assessment are limited to English text. While we cannot find any publicly available pre-trained multi-lingual sentiment classifiers, a lexicon-based method could train language-specific VADER-based lexica for the languages of interest [80]. To allow score comparability between languages, Baglini et al [81] recommend training a normalization algorithm across the included languages. However, this approach may require validation of the lexica by one or more native speakers of the languages.

A major output of this investigation is the dataset created in the process. Other, earlier studies of gender bias rely on implicit psychological techniques or sociological methods [5, 29, 30]; however, they are limited in their sample size, their sampling method (as participants are aware they are being watched), and possible researcher bias. The resulting 10 million comment Reddit dataset allows for a powerful measure of popular social gender bias. Other large studies of gender biases have relied on those present in polished media, such as news coverage and literature [22, 25, 26, 31, 33, 43]. However, these are sources where the final product is carefully manufactured to appeal to an audience and do not necessarily provide a measure of general society's biases. Many other studies using social media data exist, such as those on Twitter [16, 28, 82] or Facebook [27, 53]. However, Twitter data is limited by the character limit, and Facebook studies focus on text and language addressed *to* a gender, not *about* them, which can affect presented biases [27, 83]. Other studies on gender biases using Reddit data exist [13, 35], but this presented dataset allows the exploration of biases within politics, distinct from other gender biases. The delineation of these specific gender biases are interesting for a diverse range of subjects: computer science, linguistics, political science, and gender studies. To contribute to the uncovering of these biases, we provide the code and dataset for future use in studies.

## 6 Conclusion

In this paper, we present a comprehensive study of gender bias against women in authority on social media. We curate a dataset with 10 million Reddit comments. We investigate hostile and finer forms of bias, i.e. benevolent sexism. To this end, we employ different types of bias to assess for the nuanced nature of gender bias. We have been able to show a range of structural and text biases across two years of political commentary on the curated Reddit dataset. Though we see relatively equal public interest in male and female politicians, as measured by comment distribution and length, this interest may not be equally professional and reverent; female politicians are much more likely to be referenced using their first name and described in relation to their body, clothing and family than male politicians. Finally, we can see this disparity grow as we move further right on the political spectrum, though gender differences still appear in left-leaning subreddits.

### 6.1 Future work

Future investigations on gender biases could first match politicians on age, hierarchical level and other potentially confounding factors, by which female politicians are disproportionately affected [25]. This could also allow the investigation of other existing biases, such as those against different races or gender biases. Biases, such as racism and sexism, often interact. Elucidation of these interactions is difficult but is only possible with the use of computational techniques, such as Field et al's use of matching algorithms to show the interplay of gender, racial

and sexual biases on Wikipedia [25]. Given that all linked entities are first linked to Wikipedia pages before Wikidata IDs, a similar technique could be employed on our data, as the traits used in Field et al's matching algorithm would be accessible from the mapped Wikipedia pages.

Future investigations in combinatorial biases, nominal biases, and sentimental biases could also benefit from modifications. In-depth investigation into combinatorial biases could utilise higher-order networks (with consideration of the combinatorial issues in calculating homophily) or compute Monte Carlo simulations to investigate hypothetical causes for the observed conditional distribution. Investigations of nominal biases could be expanded to include mention of a politician's post, also a signal of respect for political authority, or to further investigate usage of nicknames, which may require hand-curated lexica of common nicknames for mentioned politicians (to avoid the accidental inclusion of misspellings). In addition, the causes of the bi-modality of the observed distribution of comment sentiment can also be further investigated. The entity-based PMI tactic could also be expanded to include document and entity-based significance measures [62]. The inclusion of these updates may lead to more nuanced results than the ones reported in this investigation.

Finally, a benefit of this dataset is its applicability for a variety of other comparisons not assessed in this study, possible thanks to the structure of Reddit. These include more cross-community comparisons, tracking user affiliations to increase the dataset size, or longitudinal comparisons. We conduct one example of a cross-community comparison with the cross-partisan comparison. However, further investigations could include inter-country comparisons, intra-generational studies (via investigation of the comments from /r/teenagers), and cross-lingual studies. Future studies could try to augment the dataset for comparisons of smaller subreddits, such as /r/MensRights or /r/feminisms. Researchers could query regular posters on these subreddits and find their posts on other larger subreddits (e.g. /r/politics, r/news). Their status as regular posters on one niche subreddit can identify them as likely belonging to one 'group', and their behaviour on larger subreddits could be added to the comparison. This carries the assumption that ones participation in a subreddit is a signal of a constant belief of that user, which may not always be applicable and must be verified. A user that once posted in a misogynist space two years ago may no longer carry misogynist views. In the case of regular posters, one could investigate how a user's behaviour may change between subreddits as a proxy to show how biases may change across communities. Lastly, time periods could be compared to assess changes in biases over time.

## Supporting information

**S1 Table. Subreddits included.**
(PDF)

**S2 Table. PMI annotations.**
(PDF)

**S1 Text. Removal of trump.**
(PDF)

## Acknowledgments

We would like to thank Dr. Jonas J. Juul for his advice and feedback in the calculation of combinatorial biases. We would also like to acknowledge those that played a role in the annotation process: Ivo, Laura, and Vivian, for their help in attributing word-senses to the terms extracted from this data-set.

## Author Contributions

**Conceptualization:** Sara Marjanovic, Isabelle Augenstein.

**Data curation:** Sara Marjanovic.

**Formal analysis:** Sara Marjanovic.

**Funding acquisition:** Isabelle Augenstein.

**Methodology:** Sara Marjanovic, Karolina Stańczak.

**Supervision:** Karolina Stańczak, Isabelle Augenstein.

**Visualization:** Sara Marjanovic.

**Writing – original draft:** Sara Marjanovic.

**Writing – review & editing:** Karolina Stańczak, Isabelle Augenstein.

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
