## [Decision Letter · Decision Letter 0]

4 Jun 2022

PONE-D-21-40358Quantifying gender biases towards politicians on RedditPLOS ONE

Dear Dr. Marjanovic,

Thank you for submitting your manuscript to PLOS ONE. After careful consideration, we feel that it has merit but does not fully meet PLOS ONE’s publication criteria as it currently stands. Therefore, we invite you to submit a revised version of the manuscript that addresses the points raised during the review process.

I apologize for the delay in submitting a decision on this manuscript. It has been very difficult to find reviewers. One expert in the field has provided a very thorough review of your paper, and I have independently read your paper. In addition to the reviewer's comments, I suggest revising the introduction. A key component of the introduction is the distinction between benevolent and hostile sexism or bias. However, in the method and results, there is no operationalization or assessment of these constructs. I have similar concerns with references made to implicit bias. It is unclear that these data can be used to assess these constructs. It may be better to discuss this content in the discussion.  

We look forward to receiving your revised manuscript.

Kind regards,

Natalie J. Shook

Academic Editor

PLOS ONE

Journal Requirements:

Reviewers' comments:

Reviewer's Responses to Questions

**Comments to the Author**

1. Is the manuscript technically sound, and do the data support the conclusions?

Reviewer #1: Yes

2. Has the statistical analysis been performed appropriately and rigorously? 

Reviewer #1: Yes

3. Have the authors made all data underlying the findings in their manuscript fully available?

Reviewer #1: Yes

4. Is the manuscript presented in an intelligible fashion and written in standard English?

Reviewer #1: Yes

5. Review Comments to the Author

Reviewer #1: I enjoyed reading this paper very much. Although the manuscript introduces data analyses that may not be accessible to the lay audience, the authors did a good job explaining in detail the analyses done. One of the attributes of these big data analyses is the effort to include different types of approaches to answer the questions outlined. This allows to derive better conclusions of the findings. The graphs and tables also support the analyses nicely and provide a good picture of what is done. I only have some minor recommendations to make the document stronger.

• Ln 14. After social media I believe authors should be written so the sentence reads better.

• Ln 21-23 I propose clarifying here that you are interested in public interest.

• Ln 43-44. Add a sentence to explain what you mean with linguistic and extra-linguistic.

• Ln 48 add a sentence clarifying that you will try to include other cultures besides the US that are represented within the English language.

• ln 49-54 clarify that you focus on public interest. Please note that this sounds repetitive with what is mentioned on ln 20-21.

• Ln 53-54 A definition is necessary for each of the 5 gender biases.

• Ln 55. At this point is distracting to read the results, especially since the results have already been presented in the abstract. I would remove this paragraph, unless there is something I am missing about the PlosOne guidelines.

• Ln71-118. The information presented here sound repetitive. I wonder if this information can be incorporated with the introduction. I recommend the authors, to establish in the introduction what is known from previous studies and when is pertinent introduce how the authors will fill the gaps in the literature. At the end of this section, the authors summarize what they are going to do, by providing additional information such as describing what the 5 gender biases they are going to analyze (if possible, here be specific about the goals in the form that is organized in the “experiments” section. Just to clarify, I am only suggesting that the goal is described in a sentence and then indicate briefly how this goal will be approached). At this point it is important to understand what the authors mean with extra-linguistic and providing the information that is presented in different sections of the paper about the language they will focus as well as the English-speaking places, besides the U.S. In short, I recommend the authors that instead of putting the goals in parts along the introduction, they should write the introduction, and the describe goals in a single section.

• ln 72 what is NLP? Since this manuscript is geared towards a wider audience, it is important that these types of terminology are fully described.

• Page 3 ln 115-118 The categories selected in this study need to be described here.

• Page 4 ln 120-122. This sentence is difficult to understand. Perhaps use two sentences?

• Ln 151-153 I don’t understand if the subreddits were not included or were purposely scrapped to be included.

• For the data section, I recommend the authors to include a table indicating which subreddits were included or excluded after scrapping the data. Perhaps even organize them as left and right. And highlighting of those scrapped, what proportion were kept for analyses. Also, if it makes sense, I recommend the authors that introduce this section by mentioning first the information of the 2 years that they focus for data collection, then mention which reddits and subreddits were scrapped, then mention that Wikidata was used to identify the target reddits that will be used in this study. The supplementary material lists the subreddits, but I think a Table, within the text that lists them as left or right and which ones were included or not included would be useful (also indicating the number of samples included within each subredit). This information will help to understand better the results.

• For the section on Experiments, I would recommend the authors to call this section Data analyses.

• 264-267 The fact that Donald Trump is such an outlier, I wonder how including this in the overall data set is skewing the results. I understand that the authors are using alternative statistics to bypass the problems with this outlier. Perhaps would be important to do analyses in a supplementary section to demonstrate that when Donald Trump data is removed from the analyses the findings remain the same. Perhaps, indicate on ln 221-229 why a different category was created for the “alt-right” group. Ideally it is important to understand in which context does women politicians are described differently. Since the “alt-right” is mostly driven by Donald Trump it makes sense that women politicians would be described differently. If relevant, this is something that can also described in the limitations.

• For results section, I wonder if the authors can do a summary after each research question. Since this manuscript needs to reach the general audience, it would be important to give summary of the take home message about the findings. Something like it is described in the sentiment analyses section.

• Ln 816 there is a type-o

• Can the authors think of ways that text analyses can be done in other languages? Perhaps include a discussion about this in the limitations, and if they can provide some solutions, would be interesting.

Minor things.

Be consistent in using terminology Men vs male

Clarify what you mean with /r/

6. PLOS authors have the option to publish the peer review history of their article (what does this mean?). If published, this will include your full peer review and any attached files.

Reviewer #1: No

---

## [Author Response · Author response to Decision Letter 0]

22 Jul 2022

Dear Editor,

We would like to thank you for collecting and relaying the review responses to us. We are resubmitting the revised manuscript with the aim to highlight the contributions of this work.

Following your concern regarding our mentions of benevolent vs hostile sexism, we have updated our several sections to explicitly define how we assess these constructs. For example, in the abstract, we now outline how benevolent sexism plays into our investigation, and our interpretation of the results. Within the Analyses section, in Ln 349 and 426-428 we note how our investigations into sentimental biases and lexical biases specifically assess for the presence of hostile or benevolent sexism. In the discussion, we specifically outline that we do not see evidence for hostile or benevolent sexism in our sentiment analyses (Lines 844-845) nor our lexical analysis (Lines 853-855). This leads to a discussion later on about we see a different, more neutral, manifestation of bias: women are more likely to be described in non-professionally relevant descriptors. We hope that these inclusions help enforce our thought-process and justify our explanations in the introduction of how language biases can take several different forms (not just hostile, but benevolent, or even neutral).

In addition, we further explain why our investigations can be justified as measurements of implicit bias in Ln 29-32 (Introduction), as the unsupervised learning techniques used in this investigation can uncover patterns in language that may not necessarily be known by the authors. In addition, previous studies in word associations have found results similar to those done in human implicit association tests. Therefore, we believe that by using unsupervised NLP tools on a large data-set, the majority of the biases we find are likely to be implicit biases.

Dear Reviewer 1,

We appreciate your feedback and review of our manuscript. We hope that you find the following response satisfactory, and that our incorporation of your recommendations helped to fortify the original document. Here are our responses to each one of your comments (noted as "Comment 1" for the first comment.

Comment 1 -- Thank you for the recommendation; we have updated Line 14 (Now Line 60 after the re-organization of the Introduction) with the names of the authors. This does make the paragraph easier to read overall.

Comment 2 -- Now that we have re-organized the introduction, we have removed these lines and instead, when we begin to outline our project, give a more specific description of what we mean by "society's biases''. In lines 72-73, we now outline that we are investigating biases in public interest as well as differences in social expectations/norms.

Comment 3 -- Thank you again for pointing out the lack of clarity in the original text. Earlier in the paragraph (Lines 52-53), we now make it clear that we are defining what we later call "extra-linguistic cues''. Then, in lines 57 to 58, we have added specific examples to ensure the reader is reminded of their meanings. We have also some sections later in the paper to consistently describe some cues as ``linguistic'' and others as "extra-linguistic'' (which were originally occasionally named textual and structural cues).

Comment 4 -- We have added a sentence in Lines 75-79 to explain how we have tried to expand the cultural relevance of this study within the English-speaking world outside of the United States.

Comment 5 -- In lines 72-73, we now outline that our investigation looks into public interest as well as differences in social expectations/norms. Given that we have re-arranged the section, we no longer have the initial mention of public interest.

Comment 6 -- We now provide a quick summary of each of the biases at the end of the introduction (Lns 86-91) after we present our contributions.

Comment 7 -- We have taken your suggestions and removed the paragraph summarizing our findings. We have re-read the Plos One guidelines and have confirmed that we do not need to include a summary of our results within the Introduction. 

Comment 8 -- After our reread, we agree that there was some redundancy within the Related Work section. Therefore, we have moved the cited work up to the relevant sections of the introduction (some were already referenced in the area). Our introduction now moves from a description of the current period, to a sociological description of biases faced by female politicians.

Then, we move into a summary of NLP and how it was been used to analyse gender bias: from supervised learning, to a history of unsupervised methods studying linguistic and extra-linguistic biases. We conclude by looking specifically at studies that have also looked at the political domain.

We then bring the reader's attention to the gaps we have presented in this research, and then highlight our contributions. We end with a description of the specific biases we investigate and a description of our partisan comparison.

Comment 9 -- We have now added a quick description of Natural Language Processing on line 16, at our first reference to a related piece of literature.

Comment 10 -- We assume by "categories'', you mean the cross-partisan analysis we investigate along the way. We have added into our description of the biases that we also compare the biases across partisanal splits of the data, to show how biases manifest differently across different communities.

Comment 11 -- Thank you. We have separated the long line into two sentences. We hope that it is now more readable and provides an adequate introduction to the section.

Comment 12 -- These subreddits were all included. We have now made this more explicitly clear in Line 153.

Comment 13 -- We have updated our S1 Table in the Appendix to include the number of comments included and the relevant partisan-affliation (if any).

We also moved the description of the time period of collection to the top. We did not scrap any subreddits. Therefore, we rewrote our description of the subreddits selected to ensure it is clear that all of these named subreddits are included in our study.

Comment 14 -- Thank you for the suggestion. We agree that the investigations we have done are not necessarily describable as ``Experiments''. However, to avoid any confusion given we already have a section named ``Data'', we have decided to simply rename the section to ``Analyses''.

Comment 15 -- Thank you for the suggestion. We re-ran all the parametric analyses (i.e. those that could be affected by such a large outlier as Donald Trump). These include the Student t-tests and ANOVAs used to compare differences in Coverage bias via comment length and Sentimental bias via comment valence and dominance score, chi-square tests and log-linear analyses used to compare differences in Nominal bias via name usage frequencies and Sentimental bias via sentiment classifier output, and $L(g_{given},g_{additional})$ calculation to determine Combinatorial bias. We confirmed that we continue to see similar patterns of results even with his removal from the data-set. We have now included a section S3 Text in the Appendix where we provide the results for the relevant analyses after the removal of Trump-related comments from the data-set. In addition, we have described in the Discussion, in lines 947-956, that, as we do not see any changes in result patterns with the removal of Donald Trump, that the biases we see are not necessarily guided by certain prominent politicians but seem to be reflective of the general ideology of posters within the communities.

Though we discuss the controversy of the /r/the\\_Donald subreddit earlier in the Data section, we also now explain that this controversy is the reason for its placement into a separate category on lines 206-207. 

Comment 16 -- We now summarize the results in 1-2 sentences in layman's terms after each question is proposed in the Results section. We hope that makes the Results and Discussion sections more readable. We debated adding some textual separation tool to make the summary more visually distinct from the rest of the section, but found they made the text less readable. Therefore, we hope their current presentation at the top of the paragraph is sufficient for the reader to follow along.

Comment 17 -- This type-o has now been resolved (``than'' has been corrected to ``that''). Thank you!

Comment 18 -- Most of the analyses are only limited by the tools available for that language. Given a suitable data-set could be created for the languages (which would require comparable NER and coreference resolution tools across the languages of interest), many of the same analyses could be extended across the other languages. Our PMI method requires the languages be one of the 21 covered languages of the spacy dependency parser. We also describe how other studies could create comparable sentiment lexica using the VADER method and a normalization function. However, a native speaker of the languages may be required to both validate the VADER-based sentiment lexica, and to annotate the gendered words from the lexical bias investigations. We now describe this in our Discussion section, within the description of the other limitations of our study.

Comment 19 -- Thanks for calling our attention to this. We have re-read the article with specific attention to the terminology. In general, throughout the article, we have tried to maintain the use of man/woman as nouns, and male/female as adjectives, following English grammatical principals.

However, given that we are looking at this through a gendered perspective (not biological), it could make sense to use man/woman as both nouns and adjectives, which is increasing in popularity, though still somewhat unconventional.

To follow convention (and to avoid redundancy given the sheer amount of mentions of men and women), we have chosen to continue using male/female as adjectives and man/woman as nouns. We have looked through the paper and made some changes where necessary to follow that rule.

Comment 20 -- For readers unfamiliar with Reddit, we have described that the prefix /r/ is used to denote a subreddit when we first introduce the concept in lines 104-105.

---

## [Decision Letter · Decision Letter 1]

26 Aug 2022

Quantifying gender biases towards politicians on Reddit

PONE-D-21-40358R1

Dear Dr. Marjanovic,

We’re pleased to inform you that your manuscript has been judged scientifically suitable for publication and will be formally accepted for publication once it meets all outstanding technical requirements.

Kind regards,

Natalie J. Shook

Academic Editor

PLOS ONE

Additional Editor Comments (optional):

Reviewers' comments:

Reviewer's Responses to Questions

**Comments to the Author**

1. If the authors have adequately addressed your comments raised in a previous round of review and you feel that this manuscript is now acceptable for publication, you may indicate that here to bypass the “Comments to the Author” section, enter your conflict of interest statement in the “Confidential to Editor” section, and submit your "Accept" recommendation.

Reviewer #1: All comments have been addressed

2. Is the manuscript technically sound, and do the data support the conclusions?

Reviewer #1: Yes

3. Has the statistical analysis been performed appropriately and rigorously? 

Reviewer #1: Yes

4. Have the authors made all data underlying the findings in their manuscript fully available?

Reviewer #1: Yes

5. Is the manuscript presented in an intelligible fashion and written in standard English?

Reviewer #1: Yes

6. Review Comments to the Author

Reviewer #1: The authors have improved the manuscript and addressed all my questions. I just would recommend them to revise their references. I think the reorganization of the references was not done when the manuscript was reorganized. For example after the reference 9 and 10 the authors include the references. 15, 16 and 38 which goes against PLOS ONE guidelines

7. PLOS authors have the option to publish the peer review history of their article (what does this mean?). If published, this will include your full peer review and any attached files.

Reviewer #1: No

---

## [Editor Report · Acceptance letter]

12 Sep 2022

PONE-D-21-40358R1 

Quantifying gender biases towards politicians on Reddit  

Dear Dr. Marjanovic:

I'm pleased to inform you that your manuscript has been deemed suitable for publication in PLOS ONE. Congratulations! Your manuscript is now with our production department. 

Kind regards, 

on behalf of

Dr. Natalie J. Shook 

Academic Editor

PLOS ONE